# *Trollius chinensis* Bunge: A Comprehensive Review of Research on Botany, Materia Medica, Ethnopharmacological Use, Phytochemistry, Pharmacology, and Quality Control

**DOI:** 10.3390/molecules29020421

**Published:** 2024-01-15

**Authors:** Lianqing He, Zhen Wang, Jiaxin Lu, Chen Qin, Jiajun He, Weichao Ren, Xiubo Liu

**Affiliations:** 1College of Pharmacy, Heilongjiang University of Chinese Medicine, Harbin 150040, China; hhelianqing@126.com (L.H.); wz870220@126.com (Z.W.); ljxlqz99188@163.com (J.L.); qc123456qc2021@163.com (C.Q.); 17304511090@163.com (J.H.); 2College of Jiamusi, Heilongjiang University of Chinese Medicine, Jiamusi 154007, China

**Keywords:** *Trollius chinensis* bunge, ethnopharmacological use, phytochemistry, pharmacology, quality control, review

## Abstract

*Trollius chinensis* Bunge, a perennial herb belonging to the Ranunculaceae family, has been extensively used in traditional Chinese medicine. Documented in the Supplements to the Compendium of Materia Medica, its medicinal properties encompass a spectrum of applications, including heat clearance, detoxification, alleviation of oral/throat sores, earaches, eye pain, cold-induced fever, and vision improvement. Furthermore, *T. chinensis* is used in clinical settings to treat upper respiratory infections, pharyngitis, tonsillitis, esoenteritis, canker, bronchitis, etc. It is mainly used to treat inflammation, such as inflammation of the upper respiratory tract and nasal mucosa. This comprehensive review explores the evolving scientific understanding of *T. chinensis*, covering facets of botany, materia medica, ethnopharmacological use, phytochemistry, pharmacology, and quality control. In particular, the chemical constituents and pharmacological research are reviewed. Polyphenols, mainly flavonoids and phenolic acids, are highly abundant among *T. chinensis* and are responsible for antiviral, antimicrobial, and antioxidant activities. The flower additionally harbors trace amounts of volatile oil, polysaccharides, and other bioactive compounds. The active ingredients of the flower have fewer side effects, and it is used in children because of its minimal side effects, which has great research potential. These findings validate the traditional uses of *T. chinensis* and lay the groundwork for further scientific exploration. The sources utilized in this study encompass Web of Science, Pubmed, CNKI site, classic monographs, Chinese Pharmacopoeia, Chinese Medicine Dictionary, and doctoral and master’s theses.

## 1. Introduction

*Trollius chinensis* Bunge, a perennial herb of the Ranunculaceae family, falls under the genus *Trollius* [1]. Widely distributed in Northern China, *T. chinensis* is recognized for its high ornamental and medicinal value [2]. Its dried flowers, known as Flos Trollii, serve as the medicinal component [3]. There are more than 20 identified species in the genus *Trollius* [4]. They are distributed mainly in the temperate and arctic regions of Asia, Europe, and North America, of which 16 are in China [5]. It usually grows in peatlands, swamps, wet meadows, and banks of reservoirs, as well as in mountain areas up to the alpine zone [6]. *T. chinensis*, with its significant ornamental and health-related compounds, is highly esteemed for applications in the food, medicine, and cosmetic industries [7]. Traditionally, the Chinese have employed *T. chinensis* for medicinal and tea purposes, dating back to the Qing Dynasty and recorded in Supplements to the Compendium of Materia Medica (Qing Dynasty) as “bitter in taste, cold in nature, non-toxic, mainly used for heat-clearing and detoxicating” [1,7]. It holds a prominent place in pharmacies, is frequently referenced in medical literature, and is listed in the Chinese Pharmacopoeia (Edition 2020) with five Chinese patent medicines.

Pharmacological tests have substantiated *T. chinensis*’s anti-inflammatory, anti-oxidant, anti-bacterial, and anti-viral properties, correlating closely with its chemical composition [8,9]. Over 100 compounds have been isolated from *Trollius* species, including flavonoids, organic acids, coumarins, alkaloids, terpenoids, and prenyl flavonoids in *T. chinensis*, boasting diverse biological activities [9,10,11]. To date, more than 100 compounds have been isolated from *Trollius* species. Phytochemical investigations of this plant have demonstrated the presence of flavonoids, organic acids, coumarins, alkaloids, terpenoids, and prenylflavonoids as main constituents of *T. chinensis* with diverse biological activities [12,13]. For instance, the flavonoid metabolite(s) Orientin and poncirin found in *T. chinensis* exhibited significant antiviral activity against parainfluenza type 3 (Para 3) [10]. Additionally, researchers have identified seventeen new labdane diterpenoid glycosides A–Q (1–17) in the dried flowers of *T. chinensis*, possessing therapeutic, antiviral, and antibacterial properties, establishing *T. chinensis* as a common anti-inflammatory drug and health tea [14,15]. The flowers have traditional uses in treating respiratory infections, pharyngitis, tonsillitis, and bronchitis in Chinese medicine [11]. The exploration of *T. chinensis* holds immense potential for novel medication research and therapeutic advancements [16]. This review article aims to provide comprehensive information and highlight the potential values associated with the development of *T. chinensis*.

## 2. Materials and Methods

Relevant literature was obtained from scientific databases such as TCMSP (https://old.tcmsp-e.com/tcmsp.php, accessed on 21 April 2023), Pubchem (https://pubchem.ncbi.nlm.nih.gov, accessed on 23 April 2023), Scientific Database of China Plant Species (http://db.kib.ac.cn, accessed on 10 April 2023), Google Scholar (https://xs.scqylaw.com, accessed on 5 April 2023), PubMed (https://pubmed.ncbi.nlm.nih.gov, accessed on 5 April 2023), Baidu Scholar (https://xueshu.baidu.com, accessed on 3 April 2023), Vip site (China Science and Technology Journal Database) (http://www.cqvip.com, accessed on 3 April 2023), and CNKI site (Chinese National Knowledge Infrastructure) (https://www.cnki.net, accessed on 3 April 2023). The most extensive collection of publicly available chemical data in the world is found on PubChem. Chemicals can be found using their names, structures, molecular formulas, and other identifiers. Discover information about biological activity, safety and toxicity, chemical and physical qualities, patents, literature citations, et al. The PubChem Compound, Substance, and Bioassay sub-databases are the three sub-databases that make up the PubChem database. TCMSP, which includes 499 Chinese herbal medicines, a total of 29,384 ingredients, 3311 targets, and 837 related diseases. TCMSP is a unique systematic pharmacology platform for Chinese herbal medicines, where we can find the relationship between drugs, targets, and diseases. This database platform provides information that includes identifying active ingredients, compounds, drug target networks, et al. [17]. The Database of China Plant Species is jointly constructed by the Kunming Institute of Botany, Chinese Academy of Sciences (KIB), the Institute of Botany, Chinese Academy of Sciences (IBS), the Wuhan Botanical Garden, Chinese Academy of Sciences (WBG), and the South China Botanical Garden, Chinese Academy of Sciences (SCBG). There are more than 31,000 species of higher plants in more than 3400 genera in more than 300 families, and the data content mainly includes standard names of plant species, basic information, systematic taxonomic information, ecological information, physiological and biochemical characteristic description information, habitat and distribution information, literature information, and other information.

TCMSP, Pubchem, and Web of Huayuan were used to find the chemical composition of *T. chinensis*. Most of the active components were obtained by searching for *T. chinensis* in TCMSP. Then, PubChem and Web of Huayuan were used to obtain and validate information related to the chemical structure of organic small molecules contained in the herb and their biological activities. The Web of China Plant Species Information Database is the primary source for the botanical collection of the genus *Trollius*. All the sites listed above are public databases and have access to the public database. The article is summarized using other websites that gather literature about the development of *T. chinensis* research. Diverse studies have been published in recent years. Therefore, a comprehensive review is necessary. This paper reviewed the research progress of *T. chinensis* from six aspects, including botany, materia medica, ethnopharmacological use, phytochemistry, pharmacology, and quality control, with the keywords of chemical constituents such as flavonoids, phenolic acids, anti-inflammatory effects, and antimicrobial effects, as well as related words such as pharmacological effects. We reviewed 350 related papers. This paper draws on over 120 articles on *T. chinensis* and documents some of the literature on chemical composition and pharmacological studies conducted from 1991 to 2023.

## 3. Botany

Based on the search results from the Chinese herbal medicine series of the Chinese herbal medicine resource dictionary [18], Flora of China (https://www.plantplus.cn/foc, accessed on 10 April 2023), Scientific Database of China Plant Species (DCP) (http://db.kib.ac.cn, accessed on 10 April 2023), and other websites, and complemented by an extensive array of references, the genus *Trollius* comprises 26 species, as detailed in Table 1.

*T. chinensis*, a perennial herb of medicinal significance, features dried flowers as its medicinal components [3,19].

The geographical distribution of *T. chinensis* mainly spans Asia, Europe, the temperate zones of North America, and the Arctic region. In China, it is located in Tibet, Yunnan, Sichuan, Qinghai, Xinjiang, Gansu, Shaanxi, Shanxi, Henan, Hebei, Liaoning, Jilin, Heilongjiang, Inner Mongolia, and Taiwan [4]. Additionally, it is prevalent in Russia (Far East, Siberia, and Central Asia), North Korea, Inner Mongolia, Sakhalin Island (Sakhalin Island), Nepal, and Northern Europe [20]. Thriving in light and moist conditions, *T. chinensis* flourishes best in deep, preferably heavy, and consistently moist soil, exhibiting resilience in full sun or partial shade. Typically growing at elevations between 1000 and 2000 m, it is frequently observed at approximately 1400 m in habitats with ample water and optimal light conditions, such as peatlands, marshes, wet meadows, reservoir banks, mountainous areas, and alpine areas (Figure 1) [21].

*T. chinensis* plants are glabrous, boasting columns reaching up to 70 cm in height (Figure 2). The stems, numbering 1–3, range from 3.5–100 cm tall, either unbranched or branched above the middle, with occasional basal or distal branching and sparse foliage featuring 2–4 leaves. Basal leaves, numbering 1–4, measure 16–36 cm in length and are characterized by long stalks, occasionally accompanied by 1–3 rosette leaves. The leaf blade is pentagonal, with dimensions of 3.8–12.5 cm, exhibiting a cordate, trilobated base; the petiole, measuring 12–30 cm, has a narrowly sheathed base. Cauline leaves mirror basal leaves, with lower leaves possessing long stalks and upper leaves being smaller, short-stalked, or sessile. The pedicel, mostly grey-green, extends 5–9 cm in length. Flowers appear solitarily terminal or in 2–3 cymes, with a diameter ranging from 3.8–5.5 cm. Sepals, numbering 6–19, measure 1.6–2.8 cm and exhibit varying colors among species, including pale purple, pale blue, white, golden yellow, yellow, or orange-yellow. The leaf blade is not green when dried and is isobovate or elliptic-obovate in shape. Petals, numbering 18–21, are narrowly linear, slightly longer than sepals or subequal to sepals apically attenuate, measuring 1.8–2.2 cm in length and 1.2–1.5 mm in width. Stamens, numerous and spirally arranged, range from 0.5–1.1 cm in length. Carpels, numbering 20–30, are sessile, and follicles are 1–1.2 cm in length and approximately 3 mm in width. Seeds are subobovoid, around 1–1.5 mm in length, black, and glossy. Flowering June–July, fruiting August–September [3,22,23].

## 4. Research on Materia Medica

*T. chinensis* has various nicknames. *T. chinensis* was recorded in the Annals of Shan Xi Traditional Chinese Medicine as Golden Pimple. It has been recorded in Wild Plants of Shan Xi under Asian *T. chinensis.* Tropaeolum majus, *T. chinensis* was recorded as a Supplement to the Compendium of Materia Medica (Thirty Years of Qianlong, 1765) by Shanxi Tong Zhi. Liao’s History is also recorded in the Annals of Wu Tai Mountain and the Sea of Humanity under Nasturtium. In Liao’s History Ying Wei Zhi, *T. chinensis* is recorded as *T. chinensis*, and The Book of Pictorial Guide of Chinese Plants calls it a globeflower [21,24]. *T. chinensis* was initially recognized as an ornamental plant. It was not until the Qing Dynasty that the medicinal value of *T. chinensis* was widely developed. The Record of Ennin’s Diary: The Record of a Pilgrimage to China in Search of the Law mentions that *T. chinensis* blooms in June and July [24]. After that, in the Yuan Dynasty, the poet Zhou Boqi used *T. chinensis* as the title of the Book of the Squire of Shangdu Poems, left heroic verses with the objects, and recorded the characteristics of the flowers of *T. chinensis* in the notes of the Book of the Squire of Shangdu Poems. In the Qing Dynasty, the origin of *T. chinensis* was recorded in the Shanxi Tong Zhi. In the Annals of Mount Wu Tai, under the name of nasturtium, *T. chinensis* was associated with miracles to record articles. The Widely Manual of Aromatic Plants describes the golden yellow color of the flower, seven petals, and two layers; the heart of the flower is also yellow; there are several flowers on one stem; and so on, describing in detail the flowering period, flowering characteristics, and other botanical characteristics of *T. chinensis* [19]. It appeared as a companion botanical drug to licorice in the description of licorice in the Bencao ZhengYao (Ming Dynasty, AD 1368–1644) but was not included in the book in its entirety [19]. *T. chinensis*’s medicinal functions were first recorded in Supplements to the Compendium of Materia Medica (Thirty Years of Qianlong, 1765) [22]. Modern character descriptions and fluorescence identification of *T. chinensis* have been included in the Chinese Pharmacopeia (1977 edition). *T. chinensis* is a traditional Mongolian medicine and not a widely used medicinal herb. Initially, its sources of medicinal herbs were mainly wild, and due to the lack of commercial supply, fewer applications, regional herbs, and relatively limited clinical applications and research, as well as the cold nature of *T. chinensis*, some potential safety and efficacy issues, and other factors, it has not been included in the Pharmacopoeia of China since 1985 [25]. The Chinese Pharmacopoeia (2020 edition) includes only five proprietary Chinese medicines: Jinlianhua Tablets, Jinlianhua Runhou Tablets, Jinlianhua Mixture, Jinlianhua Capsules, and Jinlianhua Granules [26].

## 5. Ethnopharmacological Use

### 5.1. Traditional Uses

*T. chinensis* serves as both a traditional Chinese medicine and a frequently used ethnomedicine. The herb can improve heat clearance, detoxification, alleviation of oral/throat soreness, earache, eye pain, cold-induced fever, and vision improvement [27]. Furthermore, it can effectively treat boils, poisons, and winds. The Shanhai Caozhuan briefly mentions *T. chinensis* as a remedy for boils, poisons, and all kinds of winds. Flowers are used in the Hebei Handbook of Traditional Chinese Medicine (1970) for chronic tonsillitis. *T. chinensis* is combined with Juhua and Guanaco, doubled in acute cases, or added with Yazhicao in equal parts. To treat acute otitis media, acute conjunctivitis, and other inflammatory diseases of the upper focus, *T. chinensis* and Juhua are each taken with three qian, and raw Gancao with one qian. Zhaobing Nan Fang records combining Nanshashen and Beishashen with 12 g of *T. chinensis* to promote yin and diminish fire, reducing spleen and kidney yin deficiency and inflammation caused by fire inadequacy. It is noted in the Manual of Chinese Herbal Medicine Commonly Used in Guangxi Folklore: Book I that *T. chinensis* has been utilized for alleviating eye inflammation and pain. Furthermore, *T. chinensis*, along with Wushuige and Mufurong, are recommended for treating malignant sores via compressing and pounding the affected site [19,28].

### 5.2. Current Use

In 2003, the Administration of Traditional Chinese Medicine of China announced a prescription for preventing atypical pneumonia. The prescription, *T. chinensis* Tang, combined six botanical drugs, including *T. chinensis*, to clear away heat, detoxify toxins, disperse wind, and penetrate evil spirits. This prescription had a significant effect on atypical pneumonia and is now commonly used to prevent and treat “plague”, such as the new coronavirus [19,29]. Its principal effects and clinical use for acute and chronic tonsillitis and other inflammatory conditions are recorded in the National Compendium of Chinese Herbal Medicine (1975). The pharmacological effects of *T. chinensis* are summarized in the Dictionary of Traditional Chinese Medicine (2006). To cope with the contemporary and rapidly changing lifestyle, the utilization of *T. chinensis* medicinal decoctions has diminished compared with previous times. Instead, they are now commonly consumed as patented medications—for example, Jinlianhua soft capsules and health products [30]. Moreover, the petals and stamens of *T. chinensis* are widely employed as a flavoring agent in culinary contexts, imparting a distinctive taste to salads, desserts, and beverages. Moreover, it can be used as a coloring agent, food additive, and dyeing agent [31]. It is also valued as an antioxidant component in cosmetics, including *T. chinensis* Pure Lotion and *T. chinensis* Spray. The ethnopharmacological uses of *T. chinensis* are shown in Table 2.

## 6. Phytochemistry

According to the search results of TCMSP (old.tcmsp-e.com/tcmsp.php, accessed on 21 April 2023), the Huayuan website (www.chemsrc.com, accessed on 23 April 2023), PubChem (https://pubchem.ncbi.nlm.nih.gov, accessed on 23 April 2023), and other websites combined with much of the literature review, the main components of *T. chinensis* include flavonoids, fatty acids, alkaloids, sterols, coumarins, tannins, and polysaccharides.

### 6.1. Flavonoids

Flavonoids stand out as the predominant bioactive metabolites within *Trollius chinensis* flowers. Numerous studies have substantiated the manifold advantageous biological properties of flavonoids, encompassing anti-oxidation, anti-inflammatory, anti-viral, and anti-tumor characteristics [37]. The flavonoids in *T. chinensis* consist primarily of flavone C-glycoside, flavone O-glycoside, dihydroflavone, and flavonols. Notably, flavone C-glycosides, predominantly hexose glycosides, exhibit unique stability due to a direct connection between the sugar group and the flavonoid parent nucleus via a c-c bond [4], forming a remarkably stable glycoside structure. The majority of flavone C-glycosides are situated at the flavone C-glycoside C-6 or C-8 positions, with a few occurring at the a-ring C-3 or C-4 positions. In *T. chinensis*, the flavone C-glycoside is positioned at the flavonoid A-ring C-8 positions [9]. Polyphenols, mainly flavonoids, including Orientin, Vitexin, and isoflavin, are highly abundant among *T. chinensis* and are responsible for antiviral, antimicrobial, and antioxidant activities. The flavone C-glycoside includes Orientin, Vitexin, and isodoxanthin. Notably, Orientin, Vitexin, and Orientin -2″-*O*-β-l-galactoside emerge as the most abundant flavonoids in *T. chinensis*. Vitexin and Orientin glycosyl exhibit robust inhibitory effects against influenza virus, *Staphylococcus aureus*, and epidermis [38]. In addition to flavone C-glycosides, flavone O-glycosides, such as Quercetin and Isoquercetin, are also discernible in *T. chinensis*. Noteworthy is the enhanced stability and reduced hydrolysis susceptibility of flavonoid carbosides like Orientin [39]. The therapeutic potential of these constituents extends to the treatment of age-related macular degeneration, cancer, cardiovascular disease, and skin repair following UV damage. Refer to Table 3 and Figure 3 for further details.

### 6.2. Organic Acids

The concentration of phenolic acids in *T. chinensis* surpasses only that of flavonoids. Specifically, Veratric acid stands out with a notably high concentration of 0.86–0.91 mg.g^−1^ [51]. Intriguingly, a distinct study revealed that the bioavailability of phenolic acid constituents in *T. chinensis* surpassed that of its flavonoid counterparts [52]. Organic acids in *T. chinensis* encompass both phenolic and fatty acids. Phenolic acids predominantly constitute derivatives of benzoic acid, further classified into two categories. The first category lacks a free hydroxyl group, including Veratric acid, benzonic acid, methyl veratrate, globeflower acid, etc. The second category possesses free hydroxyl groups, including vanillic acid, methyl-p-hydroxybenzoate, p-hydroxybenzonic acid, etc. [31]. *T. chinensis* houses a repertoire of 21 fatty acids, with saturated fatty acids as the primary components, and a total of 21 elements, constituting 57.95% of the detected substances. Palmitic acid and tetradecanoic acid exhibit relatively substantial content within saturated fatty acids. Additionally, nine types of unsaturated fatty acids comprise 30.35% of the total, featuring oleic acid, linoleic acid, palmitoleic acid, 3-(4-hydroxy-3-methoxybenzene) -2-acrylic acid, 3-(4-hydroxy-benzene) -2-acrylic acid, 4-phenyl-2-butenic acid, 3-phenyl-2-acrylic acid, (E) -11-eicosanoic acid, and (Z, Z, Z) -9, 12, 15-octadecanotrioleic acid [53].

Of significant note, three crucial phenolic acids—proglobeflowery acid (PA), globeflowery acid (GA), and trolloside (TS)—have been isolated from the flowers of *T. chinensis*. Pharmacological investigations have underscored their diverse biological activities, strongly correlated with the flower’s efficacy in treating respiratory infections, tonsillitis, bronchitis, and pharyngitis [54]. Refer to Table 4 and Figure 4 for detailed insights.

### 6.3. Alkaloids

Alkaloids, a prominent category of nitrogenous phytochemicals widely distributed in medicinal plants, stand out as crucial constituents in *T. chinensis*. The exploration of *T. chinensis* alkaloids remains limited, with only five of these compounds identified thus far. The principal pyrrolidine alkaloids include Senecionine and Integerrimine, the isoquinoline Trolline and Indole (R)-nitrile-methyl-3-hydroxy-oxyindole), and adenine [8,55,56,57,58]. Notably, Trolline emerges as the most abundant among these five ingredients [59]. Investigations indicate that *T. chinensis* flowers possess the highest total alkaloid content, while roots and branches exhibit the lowest concentrations. Among them, Trolline, an isoquinoline first discovered in *T. chinensis*, demonstrates significant antiviral and antibacterial activities. Refer to Table 5 and Figure 5 for detailed data.

### 6.4. Other Chemical Components

In addition to the aforementioned three primary active components, the flowers contain trace amounts of sterols, coumarins, tannins, and polysaccharides. Although these components exist in relatively low concentrations, their pharmacological effects are manifold, holding substantial potential for development. *T. chinensis* polysaccharides consist of neutral and acidic monosaccharides, predominantly comprising mannose (Man), rhamnose (Rha), galacturonic acid (GalA), glucose (Glu), galactose (Gal), arabinose (Ara), and fucose (Fuc) [60]. *T. chinensis* also harbors compounds like xantho-phyll-Epoxyde (C_40_H_56_O_3_) and trollixanthin (C_40_H_56_O_3_). The yellow pigment in *T. chinensis*, characterized as a fat-soluble pigment, exhibits remarkable stability under neutral and acidic conditions [61]. An undescribed phenolic glycoside, phenol A, isolated from *T. chinensis* flowers via spectroscopic methods, has revealed both its structural composition and pharmacological actions, including anti-inflammatory and antibacterial properties [17]. Furthermore, *T. chinensis* encompasses eight trace elements: Fe, Mg, Cu, Zn, Mn, Cr, Pb, and As. Research indicates minimal variations in Ca and Fe levels across *T. chinensis* from different regions, while more pronounced differences exist in Mn, Cu, and Zn levels [62,63]. For a comprehensive overview, consult Table 6 and Figure 6.

## 7. Pharmacological Effects

### 7.1. Antiviral Effect

A study exploring the antiviral properties of *T. chinensis* revealed that its five active components—Vitexin, Orientin, Trolline, Veratric acid, and Vitexin-2″-*O*-β-l-galorientin—exert their effects by modulating Toll-like receptors (a critical class of protein molecules associated with non-specific immunity/natural immunity). Specifically, the *T. chinensis* soft capsule demonstrated in vitro inhibition of human coronavirus OC43 replication, accomplished through the regulation of TLRs to suppress elevated expression of host cell cytokines such as IL-1B, IL-6, and IFN-a mRNA induced by viral infection. These findings substantiate the inhibitory mechanism of the *T. chinensis* soft capsule against the virus [65]. Examining 26 active components such as Rutin, Luteolin-7-*O*-glucoside, Kaempferol, Genistin, Apigenin, Scutellarin, Orientin, Daidzin, Vitexin, 3′-Hydroxy Puerarin, Puerarin, Daidzein, 3′-Methoxypuerarin, 2″-*O*-Beta-l-Galactoside, Rosmarinic acid, Progloboflowery acid, Caffeic acid, Protocatechuic acid, Ferulic acid, Veratric acid, Indirubin E, Oleracein E, Trollioside, Carbenoside I, 2″-*O*-(2‴-methyl butanol)isodangyloxanthin, 2″-*O*-(2‴-methylbutyryl) Vitexin, and glucose veratrate in *T. chinensis,* were observed to bind to the Mpro protein (2019-nCoV novel coronavirus pneumonia hydrolase Mpr0 protein) primarily through hydrogen bonds. This binding showcased Mpro protein-binding activity, affirming the potential of *T. chinensis* against novel coronaviruses [29]. Influencing pivotal anti-inflammatory and immunomodulatory targets, *T. chinensis*, when combined with multiple inflammatory and immunomodulatory pathways such as tumor necrosis factor-α (TNF-α), HIF-1, and Toll-like receptors (TLR), exhibits anti-influenza viral effects, particularly against influenza A [66]. The antiviral action of *T. chinensis* has been scrutinized through cyberpharmacology. While cyberpharmacological analyses offer valuable insights into pharmacological research, their reliance on network interactions between biomolecules and extensive databases introduces challenges related to data quality and reliability. Furthermore, the intricate nature of biological systems, limited experimental data, and the evolving understanding of drugs and targets require cautious consideration of credibility, necessitating further validation through pharmacological experiments [67].

Chicken embryos served as the medium for influenza virus cultivation, with the inhibitory effect of *T. chinensis* alcohol extract on viral proliferation in chicken embryo allantoic fluid evaluated through a chicken erythrocyte agglutination test. The results substantiated the direct inactivation of the influenza A virus by *T. chinensis* alcohol extract in vitro. In a parallel experiment involving influenza A virus inoculation into chicken embryos, the *T. chinensis* alcohol extract effectively curbed the proliferation of the virus within the embryos [68]. In a mouse model infected with influenza A (H1N1) virus, the study categorized the subjects into the control group, TGC group (*T. chinensis* crude extract gavage group), VI1~3 groups (virus infection model 1~3 groups), and VI + TGC 1~3 groups (treatment 1~3 groups), each comprising 10 mice. Notably, the aqueous extract of *T. chinensis* exhibited the potential to enhance the antiviral ability of mice. Subsequent comparative analyses validated the initial findings, establishing that aqueous extracts of *T. chinensis* augmented antiviral capacity in mice. Conversely, alcoholic extracts of *T. chinensis* directly deactivated the influenza A virus [69]. Furthermore, the aqueous extract of *T. chinensis* demonstrated potent inhibitory activity against the Cox B3 virus, achieving an inhibitory concentration of 0.318 mg/mL. The total flavonoids in this study displayed varying inhibitory activity against the respiratory syncytial virus, influenza A virus, and parainfluenza virus, with inhibitory concentrations of the viruses being 20.8 μg/mL and 11.7 μg/mL for Vitexin and Orientin, respectively [70]. Notably, 60% ethanolic extracts of *T. chinensis* and total flavonoids exhibited weak effects, with Protopanaxanthic acid among the organic acids demonstrating the weakest antiviral ability. While *T. chinensis* showed effectiveness against the influenza A virus, its impact on the influenza B virus was not significant [10,39,53]. Comparative assessments revealed that the alcoholic extract solution of *T. chinensis* soup displayed greater antiviral effects than the aqueous decoction of *T. chinensis* soup. Additionally, higher-purity *T. chinensis* soup extract exhibited a more robust inhibitory effect on the influenza virus. Specifically, 80% *T. chinensis* soup extract and secondary 95% *T. chinensis* soup extract demonstrated superior antiviral effects compared with 60% *T. chinensis* soup extract [71]. A study delved into the material basis of the UPLC-DAD-TOF/MS fingerprinting profile (ultra-performance liquid chromatography-tandem diode array detector-time-of-flight mass spectrometry) of *T. chinensis*, establishing its potential as the active agent against EV71 (enterovirus 71). The key active ingredients of *T. chinensis* in combating EV71 included Guaijaverin acid, an unidentified alkaloid, P-hydroxybenzene-malic acid, and 2″-*O*-acetyl Orientin [25]. In the broader context, *T. chinensis* flowers emerged as a valuable contributor to the anti-influenza virus activity of the overall formula, exhibiting relatively few side effects. The synergistic effect of *T. chinensis*, particularly in formulations like *T. chinensis* soup, has proven effective as a treatment for influenza virus [27,72].

In recapitulation, the findings indicate that the antiviral mechanism of *T. chinensis* predominantly revolves around impeding the virus-receptor binding process and restraining the cytokines/chemokines response. The unrefined flower extract derived from *T. chinensis* shields the host from inflammatory damage by intervening in the TLRs, encompassing TLR3, TLR4, and TLR7. This intervention leads to a reduction in the secretion of inflammatory factors, ultimately manifesting antiviral effects [73,74].

### 7.2. Antioxidant Effect

The varied pharmacological impacts of Orientin in *T. chinensis*, particularly its potent antioxidant effect, surpass those attributed to Vitexin. This discrepancy may be attributed to the structural disparity between Orientin and Vitexin. The oxidative activity of flavonoids with an o-diphenol hydroxyl group on the B-ring is notably more robust compared with those flavonoids possessing a singular phenol hydroxyl group attached to the B-ring [75].

To assess the antioxidant capacity of Orientin and Vitexin in *T. chinensis* concerning D-galactose-induced subacute senescence in mice, D-galactose was administered intraperitoneally [75]. The experimental outcomes revealed that Orientin and Bauhinia glycosides in *T. chinensis* effectively elevated the total antioxidant capacity (T-AOC), superoxide dismutase (SOD), glutathione peroxidase (GPGP), and glutathione peroxidase (GPP) in the tissues of the kidneys, livers, and brains of senescent mice. Additionally, these compounds increased SOD, glutathione peroxidase (GSH-Px), Na^+^-K^+^-ATPase, and Ca^2+^-Mg^2+^-ATPase activities in kidney, liver, and brain tissues. Notably, Orientin demonstrated superior efficacy over Oryza sativa in augmenting T-AOC activity within the organism [75]. The former mitigates impaired sodium ion transport and associated metabolic disorders [76], while the latter, elevated levels of Ca^2+^, adversely impact the cytoskeleton and membrane structure of neuronal cells, culminating in diminished stability and heightened membrane permeability, thereby contributing to the senescence process [16,75].

In contrast, the glycosides of Orientin and Vitexin pruriens act as antioxidants by positively modulating the activity of membrane transporter enzymes within tissue cells. Remarkably, Orientin exhibited greater efficacy than Vitexin in enhancing the activity of these tissue cell membrane transporter enzymes [15]. The robust antioxidant potential of Orientin, exceeding that of poncirin and further surpassing total flavonoids, has been corroborated in various studies. Both Orientin and Vitexin demonstrate the ability to scavenge superoxide anion, hydroxyl radical, and DPPH radical, effectively safeguarding the erythrocyte membrane. Specifically, Orientin displayed notable scavenging efficacy within the concentration range of 2.0–12.0 μg/mL. In contrast, Vitexin exhibited hydroxyl radical scavenging within the concentration range of 0–1.0 μg/mL, achieving maximum scavenging efficiency at 1.0 μg/mL, followed by a decline in scavenging effect with increasing Vitexin concentration [77].

The pharmacological mechanism underlying the antioxidant action of *T. chinensis* encompasses several key facets: (1) Scavenging of free radicals: The active constituents in *T. chinensis*, particularly flavonoids, exhibit potent free radical scavenging capabilities. This capacity enables the neutralization of free radicals both inside and outside the cell, thereby mitigating oxidative stress-induced damage [78]. (2) Stimulation of antioxidant enzyme activity: the active ingredients in *T. chinensis* stimulate the activity of antioxidant enzymes by stimulating the intracellular antioxidant enzymes such as superoxide dismutase, glutathione peroxidase, etc. [79]. This stimulation enhances the efficacy of the antioxidant system, fortifying cells against oxidative damage. In conclusion, *T. chinensis* safeguards cells from oxidative damage through the dual mechanisms of scavenging free radicals and enhancing antioxidant enzyme activity. These combined actions underscore the efficacy of *T. chinensis* as a potent antioxidant therapeutic agent.

### 7.3. Anti-Inflammatory Effect

The anti-inflammatory prowess of *T. chinensis* primarily targets the upper segment of the triple energizer, encompassing the area above the diaphragm within the human body. This region predominantly involves organs such as the stomach and throat, extending through the diaphragm and chest, including the heart, lungs, viscera, head, and face. Both the aqueous extract and 95% ethanol extracts of *T. chinensis* manifest robust anti-inflammatory activities. Notably, within the repertoire of compounds contained in *T. chinensis*, flavonoids such as Robinin, Quercetin, Vitexin, and Orientin exhibit heightened anti-inflammatory efficacy. Particularly, Vitexin and Orientin, due to their anti-inflammatory and soothing properties, along with peptide anti-histamine attributes, are deemed suitable for managing acute allergic skin conditions such as rash and eczema, as well as respiratory allergic diseases [80].

Current domestic research on *T. chinensis* underscores its potential in treating upper respiratory tract infectious diseases, including nasal mucosal diseases, by deploying an anti-inflammatory mechanism that engages multiple metabolites, targets, and pathways. Among the identified core targets, TNF and mitogen-activated protein kinase 1(MAPK1) take precedence, with the cancer factor pathway emerging as a pivotal route [81]. Additionally, Toll-like receptors 3, 4, and 7 (TLR3/4/7) have been proposed as promising common anti-inflammatory targets for *T. chinensis* constituents. This includes Vitexin, Orientin, Trolline, Veratric acid, and Vitexin-2″-*O*-galactoside, as discerned through the integration of network pharmacology and molecular docking techniques [82].

Respiratory inflammation, arising from diverse pathogens, microbial infections, influenza, nitroative stress, and compromised immune systems, can be effectively addressed by *T. chinensis* [83]. Its therapeutic spectrum extends beyond treating nasal mucosa inflammation to positively impacting upper respiratory infections. Leveraging data mining, an enriched analysis of the top 20 pathways linked to the targets and metabolites of *T. chinensis* in upper respiratory tract infection treatment identified quercetin as a highly probable compound. This conclusion was derived from the “metabolite-target-signaling pathway” network analysis [81]. Moreover, *T. chinensis* preparations exhibit therapeutic potential against upper respiratory tract infections by reducing serum inflammatory factors in patients. These factors include IL-8, IL-6, TNF-alpha, C-reactive protein, and procalcitonin, along with the modulation of T-cell subpopulation ratios [77,78,79,80]. Additionally, Orientin-2″-*O*-β-l-galactoside and Veratric acid have been identified for their anti-inflammatory effects [84]. In the clinical realm, the combination of amoxicillin, sodium, and potassium clavulanate has demonstrated the potential to reduce treatment duration and enhance therapeutic efficacy in children with acute tonsillitisn ratios [85,86,87,88].

In summary, *T. chinensis* harbors a repertoire of anti-inflammatory compounds, including Vitexin, Orientxin, Trolline, Veratric acid, and Vitexin-2″-*O*-galactoside. Notably, Quercetin may also contribute significantly to its anti-inflammatory activity [82,89]. Specifically, Orientin demonstrates efficacy in attenuating LPS-induced inflammation by impeding the production of inflammatory mediators and suppressing the expression of Cyclooxygenase 2 (COX-2) and Inducible nitric oxide synthase (iNOS) [90,91]. Vitexin-2″-*O*-galactoside exhibits substantial inhibitory effects on lipopolysaccharide (LPS)-induced inflammation, as evidenced by its impact on key factors such as tumor necrosis factor-α (TNF-α), interleukin-1β (IL-1β), iNOS, and COX-2 expression. Additionally, it mitigates the production of reactive oxygen species and exerts an anti-neurogenic role by inhibiting the NF-κB and extracellular signal-regulated kinase (ERK) signaling pathways, leading to anti-neuroinflammatory activity. However, the pharmacological mechanisms underlying the anti-inflammatory effects of the other components remain elusive.

### 7.4. Antitumour

Flavonoids derived from *T. chinensis* exhibit notable inhibitory effects on active cancer cells. Specifically, the total flavonoids from *T. chinensis* demonstrate the capacity to impede the proliferation of tumor cells by activating the mitochondrial pathway [92]. *T. chinensis* extracts exerted strong inhibitory effects on Leukemia K562 cells (K562), and HeL*T. chinensis* extracts manifest robust inhibitory influences on various cancer cell lines, including Leukemia K562 cells (K562), HeLa cells (He La), esophageal cancer cellsEc-109 (Ec-109), lung cancer cells NCI-H446 (NCI-H446), human non-lung cancer cells NCI-H446 (NCI-H446), human non-small cell lung cancer cell line A549 (A549), and human carcinoma cells HT-29 (HT-29), MCF-7, and HepG2, among others [92]. Moreover, the total flavonoid extract of *T. chinensis* significantly retards the growth and proliferation of MCF-7 cells. This involvement is characterized by the activation of caspase-3 and caspase-9, leading to induced cell apoptosis within a concentration range of 0.0991 to 1.5856 mg/mL [93]. Non-alcoholic fatty liver disease (NAFLD) stands as a clinical pathologic syndrome [94,95], with its incidence in China reaching a significant 29.2%, demonstrating an annual increase [96]. The complex interplay of metabolic disorders, such as dyslipidemia, hypertension, hyperglycemia, and persistent abnormalities in liver function tests, is closely associated with NAFLD [97]. Elevated lipid levels induce expression changes in HepG2 cells (hepatoma cells) [98]. In an investigation into the impact of total flavonoids from *T. chinensis* on HepG2 cell function induced by high sugar levels, it was observed that oxidative stress levels in hepatocytes and the metabolic balance of reactive oxygen species (ROS) in HepG2 cells were intricately linked to intracellular fat accumulation. The study conclusively demonstrated that total flavonoids from *T. chinensis* exhibit a specific therapeutic effect on HepG2 cells by influencing disease-associated processes. Tissue cultures were employed to compare the effects of high glucose concentrations and varying doses of total flavonoids from *T. chinensis* on HepG2 cells. The proliferative tendencies of lipid substances are directly correlated with ROS levels; higher lipid accumulation corresponds to elevated ROS levels. Elevated glucose concentrations intensified ROS levels, while total flavonoids from *T. chinensis* effectively attenuated ROS levels, thereby influencing HepG2 cells. In vitro, total flavonoids from *T. chinensis* demonstrated a capacity to reduce lipid substance accumulation, presenting a promising avenue for the improved treatment of NAFLD [96].

The ethanol extract derived from the total flavonoids of *T. chinensis* has been observed to induce apoptosis in HT-2 cells through the endogenous mitochondrial pathway. In addition, specific constituents of *T. chinensis,* namely Orientin and Vitexin, have demonstrated inhibitory effects on human esophageal cancer EC-109 cells. The apoptotic induction of EC-109 cells by both Orientin and Vitexin was found to correlate with increased drug action time and elevated drug concentrations. Significantly, Orientin surpassed Vitexin in effectively inhibiting the growth and apoptosis of EC-109 cells [99]. At the administration dose of 80 μM, Orientin demonstrated a more potent apoptotic effect on EC-109 cells compared with Vitexin at the same concentration, registering apoptotic rates of 28.03% and 12.38%, respectively, within the concentration range of 0.91 to 1.5856 mg/mL.

Elucidating the pharmacological mechanism underlying Orientin’s action, specifically in the context of esophageal cancer cells (EC-109), involves the up-regulation of P53 expression and concomitant down-regulation of Bcl-2 expression. This dual modulation positions Orientin as a prospective therapeutic agent for esophageal cancer. Utilizing the total flavonoids of *T. chinensis* as a model drug, our exploration delved into the molecular-level relationship and mechanism of these flavonoids, shedding light on their antitumor activity. A pertinent discovery was that Orientin affected HeLa, augmenting the Bax/Bcl-2 protein ratio. This manifested as an increase in Bax protein levels coupled with a decrease in Bcl-2 protein levels, thereby triggering apoptotic protease activation. Consequently, this inhibition of HeLa cell proliferation underscores the therapeutic potential of Orientin in cervical cancer treatment.

While the notable anti-tumor activity of *T. chinensis* extract is evident, the specific mechanistic intricacies remain elusive. Putatively, this metabolite’s impact on the signaling pathways within tumor cells plays a pivotal role. *T. chinensis* is observed to down-regulate anti-apoptotic genes *Bcl* and *Bcl-xL* while concurrently up-regulating pro-apoptotic genes such as *Bax, caspase-9*, and *caspase-3* at the mRNA levels. This concomitant suppression of *COX-2* gene expression in tumor cells is linked to inhibiting the proliferation of diverse tumor cell lines. The inhibitory effect extends to the HT-29 of human colon cancer cells, with *T. chinensis* flavonoids proving efficacious in restraining cell proliferation. The concentration-dependent inhibition of human non-small cell lung cancer A549 cells, the induction of apoptosis in lung cancer A549 cells, and the anti-lung cancer role demonstrated by these flavonoids underscore their potential therapeutic relevance. Moreover, the ability of *T. chinensis* flavonoids to impede the progression of K562 cells, retaining them in the Go/G1 phase, elucidates their protective role against leukemia. Additionally, beyond the total flavonoid components, the total saponins of *T. chinensis* showcase robust antitumor activity, albeit without significant advantages over other pharmaceutical agents [100].

### 7.5. Antibacterial Effect

*T. chinensis* manifests broad-spectrum bacteriostatic activity against both Gram-positive cocci and Gram-negative Bacilli, including Pseudomonas aeruginosa, *Staphylococcus aureus*, Diplococcus pneumoniae, and Shigella dysenteriae. The pivotal antibacterial constituents of *T. chinensis* are its flavonoids, notably Orientin and Vitexin [101,102,103,104]. In vitro assessments utilized Minimum Inhibitory Concentration (MIC) and Minimum Bactericidal Concentration (MBC) as benchmarks for analyzing *Escherichia coli*, *Salmonella*, *Staphylococcus aureus*, *Bacillus subtilis*, *Streptococcus mutans*, *Streptomyces*, *Rhodotorula*, *Aspergillus niger*, and *Candida albicans*. The 30% ethanolic extract of *T. chinensis* exhibited notable antibacterial efficacy, particularly inhibiting Streptococcus mutans, suggesting a potential therapeutic avenue for dental caries. *T. chinensis* total flavonoids, along with Orientin and Vitexin, exhibited notable inhibitory effects on Gram-positive cocci while demonstrating no discernible impact on Gram-negative Bacilli and fungi. Their most pronounced inhibitory activity was observed against *Staphylococcus aureus*, with the order of inhibitory strength being Orientin = Total flavonoids > Vitexin. Specifically, the lowest inhibitory and bactericidal concentrations were determined to be 0.15625 mg·mL^−1^ and 0.625 mg·mL^−1^ for Orientin and total flavonoids, respectively. Additionally, these components demonstrated considerable inhibitory activity against Streptococcus mutans, with the antibacterial efficacy ranking as Orientin > Total flavonoids > Vitexin. Notably, the lowest inhibitory concentration and bactericidal concentration of Orientin were 0.15625 mg·mL^−1^ and 0.625 mg·mL^−1^, surpassing the efficacy of Vitexin [15]. In investigations exploring the bacteriostatic activity of various *T. chinensis* preparations, the *Staphylococcus aureus* solution clarified at concentrations of 225 mg/mL for Jinlianhua Tablets, 56.25 mg/mL for Jinlianhua Jiaonang, 450 mg/mL for Jinlianhua Granules, and 56.25 mg/mL for *T. chinensis* oral solution. For *Bacillus subtilis*, clarification occurred at concentrations of 56.25 mg/mL for Jinlianhua tablets, 14.0625 mg/mL for *T. chinensis* capsule, 225 mg/mL for *T. chinensis* granules, and 28.125 mg/mL for *T. chinensis* oral solution. Notably, the *T. chinensis* oral solution displayed no inhibitory effect against *Escherichia coli*. These experiments revealed that the antibacterial activities of the four *T. chinensis* preparations followed the order of strength as *Bacillus subtilis* > *Staphylococcus aureus* > *Escherichia coli*, with varying minimum inhibitory concentrations (MICs) against *Staphylococcus aureus* and *Bacillus subtilis* for different *T. chinensis* preparations, ranked from strongest to weakest as Jinlianhua capsules, Jinlianhua mixture, Jinlianhua tablets, and Jinlianhua granules [105]. In the in vitro bacteriostatic efficacy assessment, the total flavonoids extracted from *T. chinensis* exhibited robust inhibitory effects against common pathogenic organisms, including *Staphylococcus epidermidis*, *Staphylococcus aureus*, *Escherichia coli*, *Streptococcus viridans*, *Salmonella paratyphi A*, and *Salmonella paratyphi B*. Notably, the total demonstrated considerable protective effects in *Staphylococcus aureus*-infected mice, showcasing a dose-dependent reduction in the 48-h mortality of the infected mice [106]. The yellow pigment of *T. chinensis*, composed of xantho-phyll epoxyde and trollixanthin, also displayed bacteriostatic properties, with varying degrees of inhibition against *Staphylococcus aureus*, *Bacillus subtilis*, and *Escherichia coli*, showing increased activity with escalating concentrations. Tecomin, a glucose ester of Veratric acid, exhibited effective inhibition against *Staphylococcus aureus* and Pseudomonas aeruginosa, with MICs of 0.256 and 0.128 mg/mL, respectively [107]. Progloboflowery acid has emerged as an effective treatment for Pseudomonas aeruginosa-induced inflammatory skin reactions. Inhibitory effects were observed for proglobeflowery acid, Vitexin, and Orientin against *Bacillus subtilis*, *Staphylococcus epidermidis*, *Staphylococcus aureus*, and *Micrococcus luteus*. *T. chinensis* total flavonoids, Vitexin, Orientin, and proglobeflowery acid displayed inhibitory effects on *Staphylococcus aureus* and *Staphylococcus epidermidis*, with MICs of 50 and 25 μg/mL, 100 and 25 μg/mL, 25 and 25 μg/mL, and 200 and 200 μg/mL. For *Micrococcus luteus* and *Bacillus subtilis*, the MICs were higher than 200 μg/mL [108]. In the investigation, *T. chinensis* extract and its three metabolites exhibited potent inhibitory effects on four Gram-positive cocci. Total flavonoids and Vitexin, having the highest content, demonstrated strong inhibition, especially Orientin, against *Staphylococcus aureus* and *Staphylococcus epidermidis*, while PA demonstrated relatively weak inhibition against these two bacteria [100,106]. The study further revealed that PA had robust inhibitory action against Pseudomonas aeruginosa and *Staphylococcus aureus*, with MIC rates of 16 and 200 mg/L, respectively. Additionally, PA exhibited modest antiviral activity (IC50 of 184.2 μg/mL) against Para 3. Conversely, GA displayed significant antiviral efficacy against influenza A, as evidenced by its IC50 value of 42.1 μg/mL. With a MIC rate of 128 mg/L, TS demonstrated moderate inhibitory activity against Streptococcus pneumonia [43].

The antibacterial pharmacological mechanism underlying the action of *T. chinensis* predominantly revolves around impeding regular bacterial growth processes. This is accomplished by elevating extracellular nucleic acid and soluble protein levels within bacteria. The resultant damage to the cell membrane influences membrane permeability, inducing the efflux of vital metabolic substances crucial for cellular viability or the influx of detrimental medicinal fluids. Such interactions significantly impact bacterial growth, thereby realizing the intended inhibitory effects. The drug concentration exhibits a positive correlation with both the rate of inhibition of bacterial growth and the rate of inhibition of biofilm formation [105,109].

### 7.6. Others

The main active components of *T. chinensis*, total flavonoids, also have analgesic and antipyretic effects. Studies have shown that flavonoids can significantly reduce ET (the lipid and polysaccharide substances produced by the cell wall of G-bacteria-ET are a standard model for screening antipyretic drugs and exploring antipyretic mechanisms). Total flavonoids can also reduce the contents of endogenous heat sources TNF-α and IL-1β in the serum of febrile rabbits and then inhibit the production and release of PGE2 in the cerebrospinal fluid of rabbits by inhibiting the production or release of TNF-α and IL-1β induced by ET to reduce fever, increase heat loss, and restore body temperature to normal. Reducing the production of endogenous pyrogens such as IL-1 and TNF-α is the pharmacological basis of the antipyretic effect of total flavonoids [110].

The experiment was divided into two parts: the blank group, the positive group, the water extract from stem and leaf (low), the water extract from stem and leaf (high), the alcohol extract from stem and leaf (low), and the alcohol extract from stem and leaf (high). The control group was used to investigate the anti-inflammatory effect of *T. chinensis*. A part of the control group was divided into the blank group (distilled water 20 mL/kg), positive group (100 mg/kg), low (12 g/kg), and high (24 g/kg) water extract groups, and low (12 g/kg) and high (24 g/kg) alcohol extract groups as the control group to verify the analgesic effect of *T. chinensis*. The extracts of *T. chinensis* stem and leaf have anti-inflammatory and analgesic effects [111]. One study further investigated the antitussive, anti-inflammatory, and analgesic effects of *T. chinensis* [112]. The study showed that all dose groups of total flavonoid extract of *T. chinensis* could significantly prolong the therapeutic effect of antitussive; with the increase in total flavonoid extract dose, the incubation period of cough in mice was prolonged, and the number of coughs was reduced—the more significant the tracheal phenol red excretion, the more pronounced the antitussive and expectorant effects. The antitussive and expectorant effects were more evident in the high-dose group of total flavone extract of *T. chinensis* than in patent medicine cough syrup. In addition, the high-dose group treated with the whole flavonoid extract of *T. chinensis* showed significantly reduced ear swelling caused by xylenes, reduced reaction times, and an improved hot plate pain threshold.

Moreover, statistical analysis showed that the total flavonoid extract of this drug had effects similar to those of nonsteroidal anti-inflammatory drugs commonly used in clinics. It was confirmed that the total flavone extract had sound anti-inflammatory and analgesic effects and that the total flavones of *T. chinensis* were helpful in myocardial ischemia-reperfusion injury. Experimental studies have shown that its mechanism of action is to inhibit the activities of superoxide dismutase (SOD) and glutathione peroxidase (GSH-Px), reduce the content of malondialdehyde (MDA), reduce the area of myocardial infarction, inhibit the release of myocardial enzymes, reduce the apoptosis of myocardial cells, and play a corresponding therapeutic and relieving role [113]. In addition, Orientin and Vitexin in *T. chinensis* could improve membrane transport in d-galactose-induced aging mice, which may be helpful for clinical applications in treating acute respiratory distress syndrome [102,114] (Table 7, Figure 7).

## 8. Quality Control

### 8.1. Analysis Methods

Currently, the market for Chinese herbal medicine *T. chinensis* has not been unified into varieties, in addition to *Trollius chinensis* Bunge. as the primary source of medicinal botanical drugs, *Trollius ledebourii* Reichenbach. *Trollius macropetalus* Fr. et al. have also done more research on resource exploitation and utilization for medicinal use. Hence, the quality of *T. chinensis* on the market is confusing, and it is difficult to distinguish the good from the bad. The 1977 edition of the Chinese Pharmacopoeia analyzes the quality of botanical drugs from two perspectives: physical identification and chemical identification. The 1998 edition of the Beijing Standards for Chinese Materia Medica (1998) also includes a microscopic identification method for determining authenticity. The 2019 edition of the Anhui Provincial Standard for the Preparation of Chinese Medicinal Tablets (2019) records the method of identification by thin-layer chromatography, in which the chromatograms of the test article obtained by experimental treatment and the chromatogram of the control botanical drug show spots of the same color at the corresponding positions of the thin-layer plate. The evaluation method in the 2018 edition of the Hubei Quality Standard for Traditional Chinese Medicinal Materials (2018) specifies that the moisture content of *T. chinensis* should not exceed 13.0%. The total ash content should not exceed 9.0%. The leachate content shall not be less than 35.0%. The content of Orientin (C_21_H_20_O_11_) must not be less than 1.0% when measured by high-performance liquid chromatography and calculated on the dry product. In addition to the identification methods recorded in pharmacopeia and local standards, the fluorescence reaction identification method, micro-sublimation test, FTIR identification, and DNA barcode molecular identification method of Chinese herbal medicines can also be used to identify the authenticity of *T. chinensis*. A micro-sublimation test can be seen on the slide of yellowish snow-like crystals [122]. The FTIR profile of *T. chinensis* was obtained by using FTIR identification, and the differences in peak shape, peak position, and peak intensity of the peaks in the profile can elucidate the differences in the components, compositions, and ratios of *T. chinensis* botanical drugs extracted from different origins, habitats, varieties, growth years, and different drying methods and extraction solvents to carry out a more accurate quality analysis to determine the authenticity of *T. chinensis* [123,124,125]. DNA barcode molecular identification of Chinese herbal medicines is a method to identify herbal medicines through the study of the polymorphism of the genetic material of Chinese herbal medicines, which can quickly identify the species [126]. At present, with the rapid development of molecular identification technology and in-depth plant genetic information mining, molecular identification methods in the standardization of traditional Chinese medicine identification have been widely used. For example, the early DNA molecular identification technique of *T. chinensis*, random amplified polymorphic DNA labeling (RAPD), was used to identify *T. chinensis* by observing the electrophoretic results of the DNA bands by PCR amplification, and the samples of *T. chinensis* could be classified according to their origins by using the RAPD technique [101]. The DNA barcode identification method of *T. chinensis* was established by using ITS2 sequences, and the neighbor-joining (NJ) phylogenetic tree was constructed to accurately identify *T. chinensis*, *Trollius lilacinus* Bunge, and *Artemisia annua* L. In addition, high-performance liquid chromatography (HPLC) coupled with mass spectrometry (MS) can be used to identify the chemical composition and characteristics of TCM. Based on the different information of protein bands of different varieties as the basis for the identification of TCM with protein as the informative substance, it is observed that the protein bands of different varieties of *T. chinensis* differ significantly in the number of bands, levels, and distribution [127]. In addition to the above methods, X-ray diffraction and X-ray fluorescence analysis can also be used to identify the grain characteristics of *T. chinensis* and establish a primary X-ray diffraction database for rapid identification of the authenticity of *T. chinensis* and its powder [128].

### 8.2. Quality Evaluation Method

To ensure the quality and therapeutic efficacy of *T. chinensis*, the key to quality control of the active ingredients is also to establish quality analysis methods. The quality of *T. chinensis* can be identified and evaluated through the establishment of content determination standards, the use of fingerprinting evaluation methods, and other methods that can provide reference for the further development and utilization of *T. chinensis*. At present, the quality evaluation method of *T. chinensis* is mainly based on chemical content determination, i.e., HPLC fingerprinting, with Orientin and Vitexin as the index components of the method [129,130]. In some studies, these two metabolites are combined with phenolic acid or alkaloid and other metabolites as quality evaluation indexes to improve the comprehensiveness of evaluation, and HPLC is the main evaluation method at present [8,39,59,131].

## 9. Conclusions and Future Perspectives

Based on ancient texts and modern research, this paper reviews the herbal testimonies, traditional uses, phytochemistry, pharmacological activities, and quality standards of *T. chinensis* to provide new ideas for future research on *T. chinensis*. According to ancient texts, *T. chinensis* can reduce inflammation, eliminate heat and toxins, and enhance visual clarity. It is particularly effective in managing sore throats, swollen gums, and oral gingival pain caused by heat. Based on recent phytochemical and pharmacological studies, *T. chinensis* possesses anti-inflammatory, antiviral, antitumor, antibacterial, and antimicrobial effects, which are especially good for treating virus-induced colds and various types of inflammation, such as respiratory inflammation. It was initially recorded as an ornamental plant in various ancient books. Since its initial inclusion in the Compendium of Materia Medica as a traditional Chinese medicine in 1765 during the Qing Dynasty, *T. chinensis* has been widely developed for its medicinal properties and employed in health care products and various dosage forms following current processing technology. Over 180 compounds from *T. chinensis* have been isolated and identified. The main active components of *T. chinensis* are flavonoids, alkaloids, and organic acids. Objective evaluations are emphasized in recent studies of *T. chinensis*, where the focus is mainly on the flavonoids Orientin and Vitexin. These two compounds are the most important and representative of *T. chinensis*, with less research on the other active components. Various domestic and international investigations indicate that flavonoids account for most of the pharmacological effects of *T. chinensis*.

First of all, regarding the medicinal employment of *T. chinensis*, historical records specify that its dried flower is the primary constituent. Additionally, contemporary experimental research concentrates on the flower of *T. chinensis*; however, chemical makeup and pharmacology evaluations of its roots, stems, and leaves are limited. Moreover, most research on the phytochemical metabolites of *T. chinensis* concentrates on crude extracts and flavonoids, including Vitexin, Orientin, and Orientin-2″-*O*-β-l-galactopyranoside. However, there is a lack of studies on the alkaloids and organic acids present in *T. chinensis,* with only a limited number of articles on this topic.

Second, studies have shown that both crude extracts and active constituents of *T. chinensis* have a wide range of pharmacological activities, and these modern pharmacological studies support most of the traditional uses of *T. chinensis* as a folk medicine. However, there is still a gap in the systematic research on *T. chinensis*. Many pharmacological studies on its crude extracts or active constituents are not in-depth enough, and fewer in vitro experiments exist. These pharmacological activities must be further confirmed by in vivo animal experiments and combined with clinical applications. This direction will provide a solid foundation for developing novel drug-lead compounds. For example, relevant animal experiments did not verify the antitumor effect of *T. chinensis*.

Third, most studies on the pharmacological activities of *T. chinensis* have focused on uncharacterized crude extracts, making it difficult to clarify the link between the isolated compounds and their biological activities. Systematic pharmacological studies on compounds isolated from *T. chinensis* are considerable. In addition, many pharmacological activities of crude extracts or compounds of *T. chinensis*, such as the anti-inflammatory pharmacological effects of *T. chinensis,* are currently focused on network pharmacology and molecular docking techniques, with only very few relevant in vitro experiments for further validation, and the exact mechanism of the inhibitory activity is still unclear; therefore, further studies to reveal better the precise molecular mechanism of the pharmacological activity of the drug appear to be necessary.

Fourth, in some ancient texts, *T. chinensis* was used with other botanical drugs, thereby treating chronic inflammation. However, almost no studies have been carried out to investigate the formulae of *T. chinensis* or to reveal the effects of synergistic or antagonistic actions. The area of this piece is almost blank. Therefore, drug interactions between certain botanical drugs and *T. chinensis* seem to be a new direction worth further exploration.

Fifth, *T. chinensis* was included in the 1977 edition of the Chinese Pharmacopoeia, but this variety was not included in the 1985–2020 edition. Although this paper summarizes the identification methods of *T. chinensis* in other pharmacopeias, the provisions on authenticity identification and quality evaluation methods of *T. chinensis* are not comprehensive compared with other Chinese medicinal materials. For example, *Trollius ledebourii* Rchb. It is an alternative source of *T. chinensis*. However, the different base plants of *T. chinensis* have not been included in the pharmacopeia like other Chinese botanical drugs, which limits the further development and utilization of *T. chinensis*. In addition, although other plants of the same genus have been used as substitutes for *T. chinensis* in some places, there is no unified standard in the market for evaluation, confusing product types, specifications, and grades of Chinese medicinal materials in the medicinal materials market, which easily leads to problems in efficacy and safety. At present, the commonly used identification methods for *T. chinensis* are different. Microscopic identification and character identification make it difficult to distinguish the difference between *T. chinensis* and different species of *T. chinensis*. Molecular identification technology still needs to be further improved, and new DNA molecular marker technology must be developed. By analyzing and comparing the ribosomal DNA of biological species, species identification methods such as ITS barcode technology still need to collect more *T. chinensis* from different places and species to improve relevant studies and further verify the applicability of this method.

In summary, *T. chinensis* serves not only as an ornamental plant and a tea source but also as a significant medicinal and food crop, possessing wide-ranging pharmacological and nutritional value. Nonetheless, more in-depth and comprehensive clinical utility studies are needed to establish the plant’s safety and effectiveness. Various compounds have been identified in *T. chinensis*, although the work done so far has been insufficient. Furthermore, additional research is necessary to determine the precise molecular mechanisms of these active ingredients in specific diseases. Future investigations should emphasize active metabolites other than flavonoids to uncover novel compounds and pharmacological effects. Thus, systematic studies on the phytochemistry and bioactivity of *T. chinensis* are essential for future research endeavors. This review is intended to serve as a valuable reference for developing and applying *T. chinensis.*

## Figures and Tables

**Figure 1 molecules-29-00421-f001:**
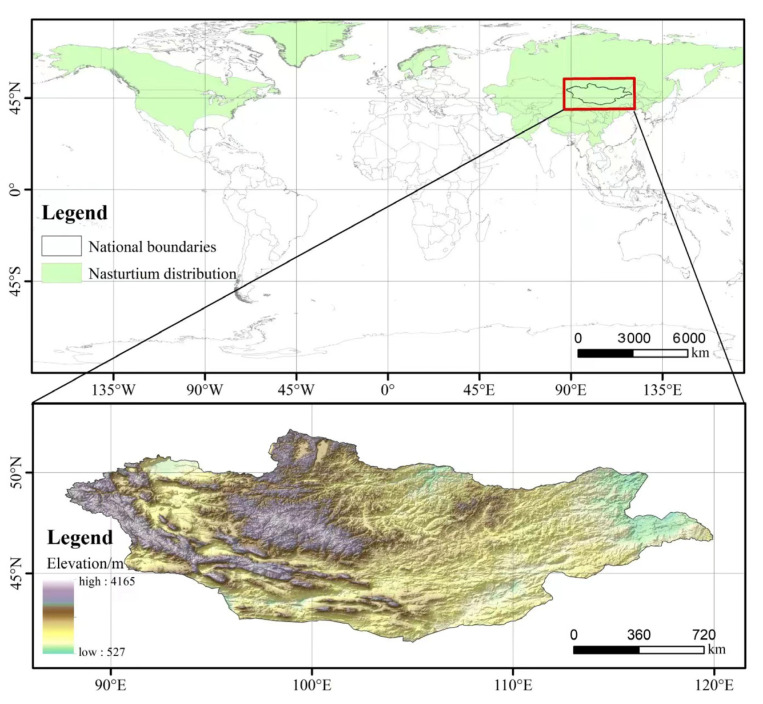
Distribution of *T. chinensis.* (The green shading represents the distribution of *T. chinensis*, white is the area where *T. chinensis* almost does not exist, and the bottom half of the image is the typical height legend of *T. chinensis* (map approval number: GS(2019)1822)).

**Figure 2 molecules-29-00421-f002:**
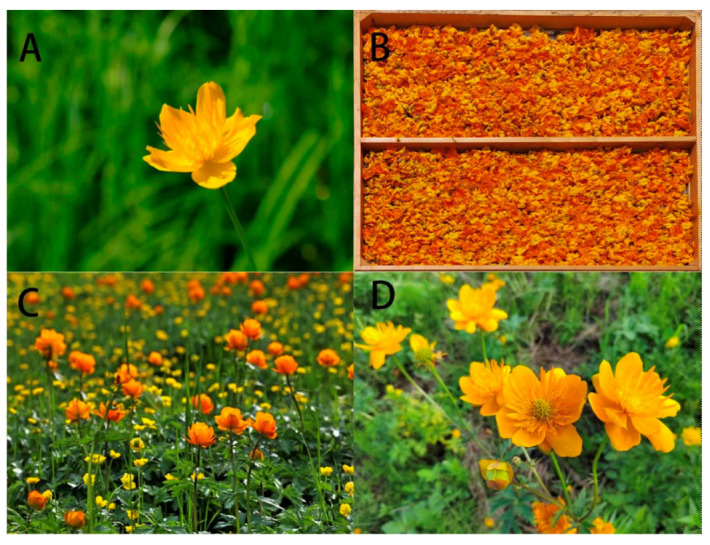
The plant of *Trollius chinensis* Bunge (Ranunculaceae). (**A**,**C**,**D**) flowers (**B**) the medicinal part after processing.

**Figure 3 molecules-29-00421-f003:**
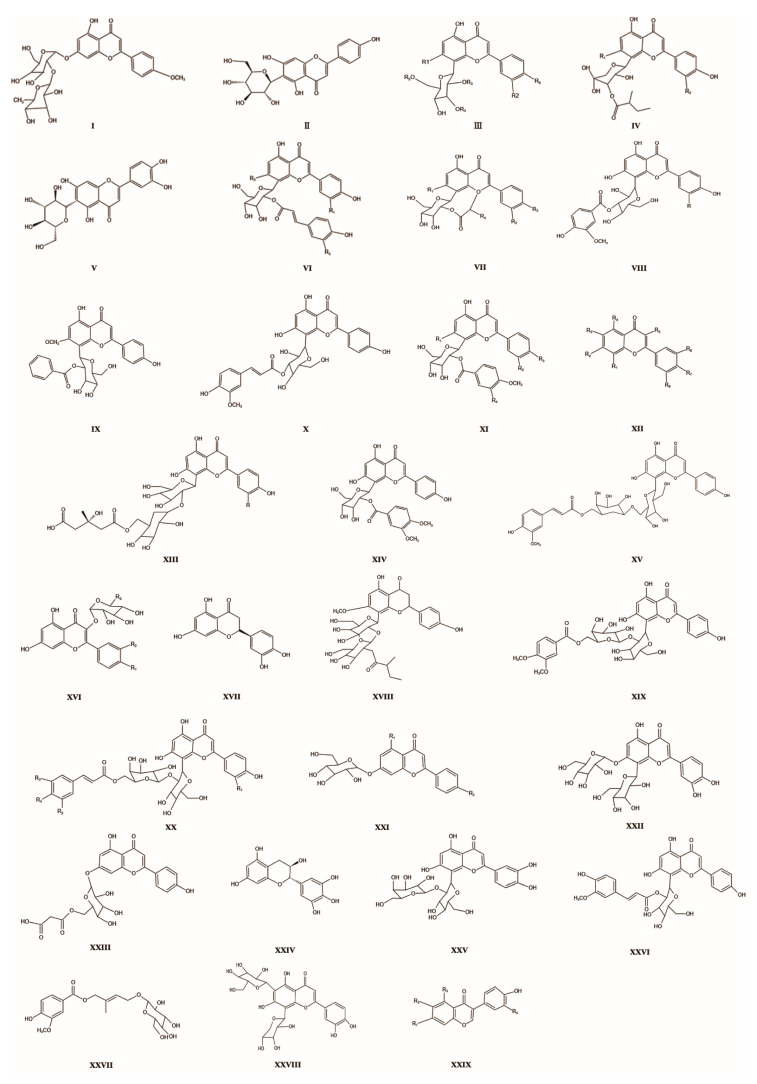
Parent nucleus structure of flavonoid chemicals in *T. chinensis*.

**Figure 4 molecules-29-00421-f004:**
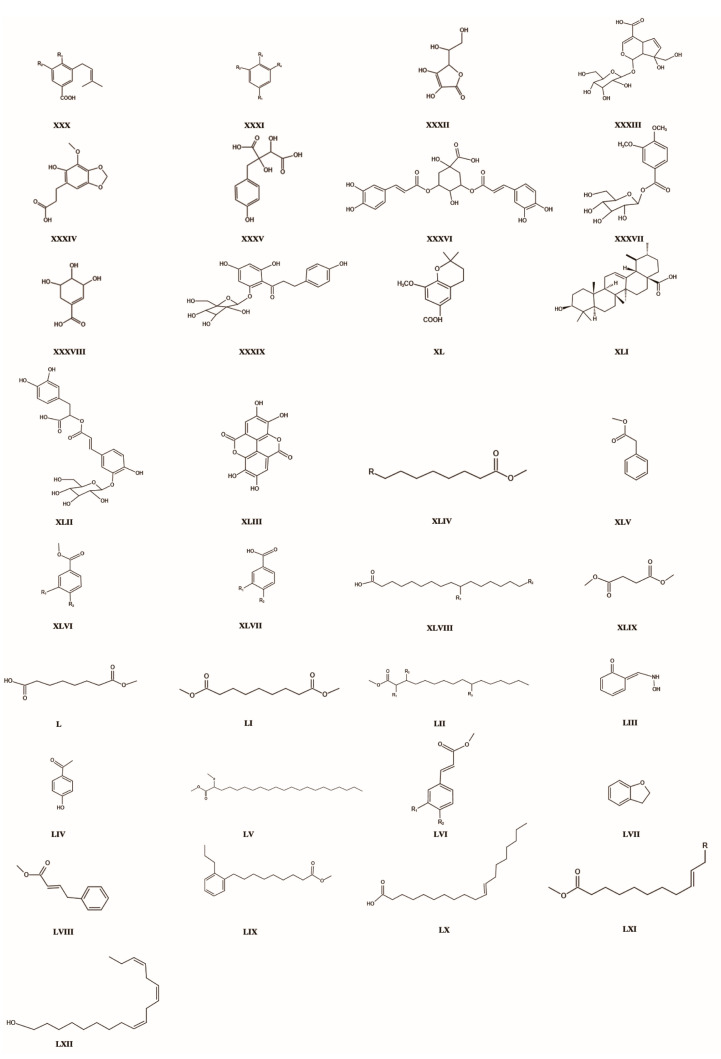
Parent nucleus structure of Phenolic acids chemicals in *T. chinensis*.

**Figure 5 molecules-29-00421-f005:**
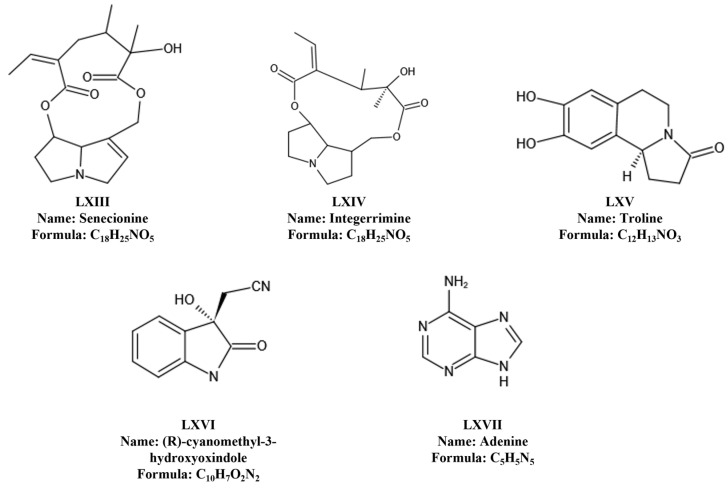
Alkaloids isolated from *T. chinensis*.

**Figure 6 molecules-29-00421-f006:**
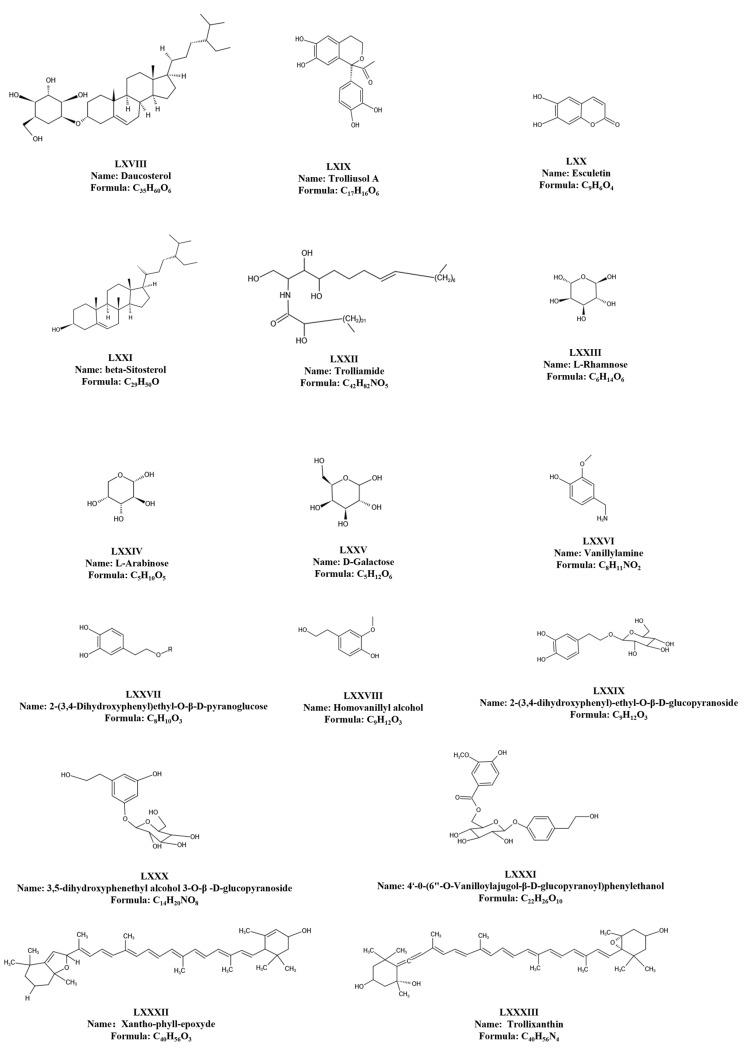
Other chemical components isolated from *T. chinensis*.

**Figure 7 molecules-29-00421-f007:**
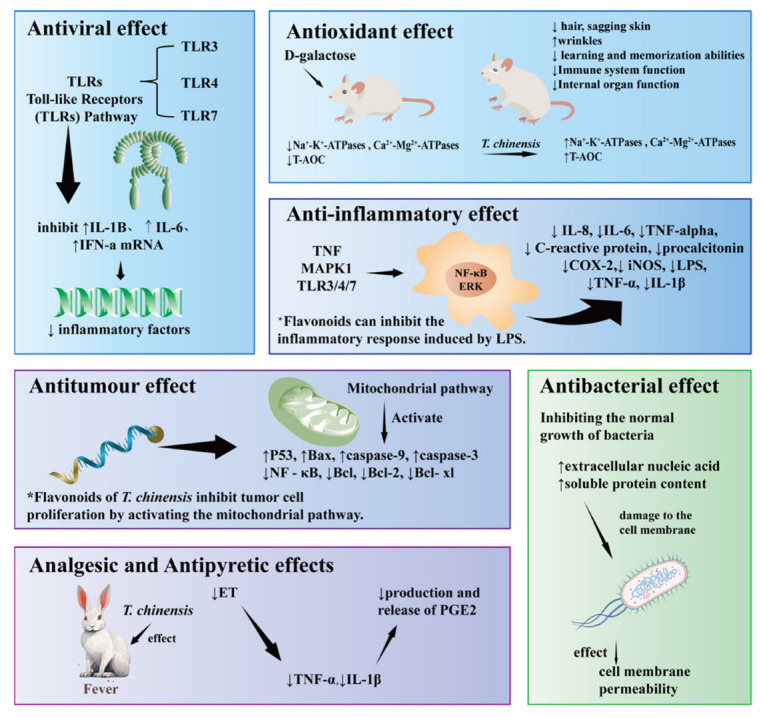
The pharmacological properties of *T. chinensis*.

**Table 1 molecules-29-00421-t001:** A total of 26 species of *Trollius* genus.

Num	Latin Name	Distribution Area	Altitude
1	*T. chinensis* Bunge	Shanxi, N. Henan, Hebei, E. Inner Mongolia, W. Liaoning and Jilin provinces of China	1000–2200 m
2	*Trollius altaicus* C. A. Mey.	N. Xinjiang (Tacheng, Altai, etc.), China; W. Inner Mongolia, China; Siberia, Russia; People’s Republic of Mongolia	1200–2650 m
3	*Trollius asiaticus* L.	Heilongjiang, China (Shangzhi); Xinjiang, China (Hami); Siberia, Russia; Mongolia	Not applicable
4	*Trollius buddae* Schipcz.	N. Sichuan, China; S. Gansu, China; S. Shaanxi, China	1780–2400 m
5	*Trollius buddae* f. *dolichopetalus* P. L. Liu and C. Du	Not applicable	Not applicable
6	*Trollius dschungaricus* Regel	Tianshan and Zhaosu, Xinjiang, China; Central Asia, Russia	1800–3100 m
7	*Trollius farreri* Stapf	Qinghai, China	2000–4700 m
8	*Trollius farreri* Stapf var/. major W. T. Wang	NW Yunnan, China (Deqin); SE Tibet, China (Tsatsumi)	3500–4200 m
9	*Trollius japonicus* Miq.	Changbai Mountain, Jilin, China; Sakhalin Island (Kuril Islands); Japan	1200–2300 m
10	*Trollius ledebourii* Rchb.	Heilongjiang, China; NE Inner Mongolia, China; E. Siberia, Russia; Far East	110–900 m
11	*Trollius macropetalus* Fr.	Liaoning, China; Jilin, China; Heilongjiang, China, etc.; Russian Far East; N Korea;	450–600 m;
12	*Trollius micranthus* Hand.-Mazz.	NW Yunnan (Deqin); E. Tibet (Motuo)	3900–4200 m
13	*Trollius pumilus* D. Don	Southern Tibet, China; Nepal; Sikkim	4100–4800 m
14	*Trollius pumilus* D. Don var. foliosus (W. T. Wang) W. T. Wang	Min County, S. Gansu, China	3000–3400 m
15	*Trollius pumilus* D. Don var. tanguticus Brühl	NE Tibet, China; NW Sichuan, China; S. and E. Qinghai, China; SW Gansu, China.	2300–3700 m
16	*Trollius pumilus* D. Don var. tehkehensis (W. T. Wang) W. T. Wang	Dege, Sichuan, China	Not applicable
17	*Trollius ranunculoides* Hemsl.	NW Yunnan, E Xizang, W Sichuan, S and E Qinghai, S Gansu, China.	2900–4100 m
18	*Trollius taihasenzanensis* Masam.	Taiwan, China	3400–3900 m
19	*Trollius vaginatus* Hand.-Mazz.	NW Yunnan (Zhongdian), China; SW Sichuan (Muli), China.	3000–4200 m
20	*Trollius yunnanensis* (Franch.) Ulbr.	W. and NW Yunnan, China; W. Sichuan, China.	2700–3600 m
21	*Trollius yunnanensis* (Franch.) Ulbr. var. anemonifolius (Brühl) W. T. Wang	W. Sichuan and S. Gansu, China.	3050–3800 m
22	*Trollius yunnanensis* (Franch.) Ulbr. var. eupetalus (Stapf) W. T. Wang	Gonshan and Deqin, NW Yunnan, Sichuan, China	3300–3900 m
23	*Trollius yunnanensis* (Franch.) Ulbr. var. peltatus W. T. Wang	Emei area, Sichuan, China	1900 m
24	*Trollius lilacinus* Bunge	Tian Shan, Xinjiang, China; W. Siberia, USSR; Central Asia	2600–3500 m
25	*Trollius laxus*	the United States in Conn.(Connecticut), Del.(Delaware)NJ.(New Jersey)N.Y.(New York)Pa.(Pennsylvania), Ohio.(Ohio)	Not applicable
26	*Trollius europaeus*	N. Europe, Central Europe and W. Asia	Not applicable

Not applicable means that no relevant information is to be found.

**Table 2 molecules-29-00421-t002:** The ethnopharmacological use of *T. chinensis*.

NO	Ethnopharmacological Use	References
1	Ornamental: The whole flower is golden yellow. It blooms in June. In autumn, the flowers are dry, and the fruit is like millet.	Guang Qun Fang Pu Kangxi (AD. 1708) [32]
2	Drink: Dry long-term preservation, to spend some tea, a pot of one, boiled water. Medicinal use: taste smooth and bitter, non-toxic, cold, cure sore throat, heat flotation tooth declaration, ear pain, eye pain, and fry this generation of Ming.	A sea record of Cha Shenxing Kangxi (AD. 1713) [33]
3	Medicinal use: bitter taste, cold, non-toxic, treating mouth sore throat swelling, ear pain, eye pain, sore throat, fever from a cold, eyesight.	Bencao Gangmu Shiyi: A Supplement to Compendium of Materia Medica (AD. 1765) [22]
4	Medicinal use: treatment of furunculosis big poison, bias wind, wind heat, wind hysteria, and wind arthralgia, et al.	Mountain sea grass letter (Qing dynasty) [22]
5	Medicinal use: clearing away heat and toxic materials, treatment of chronic/acute tonsillitis, acute otitis media, acute tympanitis, acute conjunctivitis, and acute lymphangitis.	Hebei Traditional Chinese Medicine Manual (1970) [34]
6	Medicinal use: for treating blade wounds and pulse wound sores; for swollen lymph glands and sore throats.	Compilation of Mongolian medical formulas (In 2004) [35]
7	Medicinal use: cure fever from an ear infection or eye disease.	Inner Mongolia Herbal Medicine (1972) [36]
8	Jinlianhua Mixture: clearing heat and removing toxins for upper respiratory tract infections, pharyngitis, and tonsillitis.	2020 Edition of Chinese Pharmacopoeia (CP) (2020) [26]
9	Jinlianhua tablets: clearing heat and removing toxins for upper respiratory tract infections, pharyngitis, and tonsillitis.	2020 Edition of Chinese Pharmacopoeia (CP) (2020) [26]
10	Jinlianhua capsules: Clearing heat and removing toxins, relieving pharynx and swelling. Suitable for treating inflammation of the upper Jiao, etc.	2020 Edition of Chinese Pharmacopoeia (CP) (2020) [26]
11	Jinlianhua granules: Treats upper respiratory tract infections, pharyngitis, and tonsillitis. Relieves inflammation and pain.	2020 Edition of Chinese Pharmacopoeia (CP) (2020) [26]
12	Jinlianhua Runhou tablets: clearing heat, removing toxins, reducing swelling, relieving pain, and improving the taste of the throat.	2020 Edition of Chinese Pharmacopoeia (CP) (2020) [26]
13	Jinlianhua granules: It is effective in clearing heat and removing toxins, promoting the production of body fluids, improving the pharynx, and relieving cough and expectoration. It is suitable for symptoms of heat and toxicity caused by colds, including high fever, thirst, and dry throat, and for the above symptoms caused by influenza and upper respiratory tract infections.	2020 Edition of Chinese Pharmacopoeia (CP) (2020) [26]

This table summarizes the uses and origins of *T. chinensis* from ancient to modern times.

**Table 3 molecules-29-00421-t003:** Flavones isolated from *T. chinensis*.

No	Names	Molecular Formula	Parent Nucleus	Substituent	CAS	Molecular Weight	Refs.
1	3″-*O*-Acetylquercetin	C_28_H_32_O_14_	I	nothing	nothing	592.50	[8]
2	Isorhamnetin	C_21_H_20_O_10_	II	nothing	480-19-3	432.38	[40]
3	Icariin	C_21_H_20_O_10_	III	R_1_ = OH; R_2_ = H; R_3_ = H; R_4_ = H; R_5_ = H; R_6_ = OH	489-32-7	432.38	[8]
4	Apigenin	C_21_H_20_O_11_	III	R_1_ = OH; R_2_ = OH; R_3_ = H; R_4_ = H; R_5_ = H; R_6_ = OH	520-36-5	448.41	[40]
5	Isoswertisin	C_22_H_22_O_10_	III	R_1_ = OCH_3_; R_2_ = H; R_3_ = H; R_4_ = H; R_5_ = H; R_6_ = OH	6980-40-1	446.40	[40]
6	Isoswertiajaponin	C_22_H_22_O_11_	III	R_1_ = OCH_3_; R_2_ = OH; R_3_ = H; R_4_ = H; R_5_ = H; R_6_ = OH	nothing	462.40	[8]
7	Trollisin I	C_22_H_22_O_10_	III	R_1_ = OCH_3_; R_2_ = H; R_3_ = H; R_4_ = H; R_5_ = H; R_6_ = OH	nothing	446.40	[8]
8	Cyanidin 2″-*O*-(β-d-xyranosyl)-β-d-glucoside	C_26_H_28_O_16_	III	R_1_ = OH; R_2_ = OH; R_3_ = D-xyl; R_4_ = H; R_5_ = H; R_6_ = OH	nothing	596.49	[8]
9	Cyanidin 2″-*O*-(β-d-pyranosyl)-β-d-glucoside	C_26_H_27_O_15_N	III	R_1_ = OH; R_2_ = OH; R_3_ = D-glu; R_4_ = H; R_5_ = H; R_6_ = OH	nothing	593.50	[8]
10	Cyanidin 2-prime-*O*-beta-pyranosyl-arabinoside	C_26_H_28_O_16_	III	R_1_ = OH; R_2_ = OH; R_3_ = D-ara; R_4_ = H; R_5_ = H; R_6_ = OH	nothing	596.50	[8]
11	Cyanidin 2″-*O*-beta-l-rhamnoside	C_27_H_30_O_16_	III	R_1_ = OH; R_2_ = OH; R_3_ = L-gal; R_4_ = H; R_5_ = H; R_6_ = OH	nothing	609.15	[8]
12	Cyanidin 3-*O*-beta-d-glucoside-6″-*O*-alpha-l-rhamnoside	C_26_H_27_O_15_N	III	R_1_ = OH; R_2_ = OH; R_3_ = H; R_4_ = H; R_5_ = D-glu; R_6_ = OH	nothing	593.50	[8]
13	6″-*O*-Acetyl cyanidin	C_32_H_27_O_11_N_3_	III	R_1_ = OH; R_2_ = OH; R_3_ = H; R_4_ = H; R_5_ = Ac; R_6_ = OH	nothing	629.58	[8]
14	3″-*O*-Acetyl cyanidin	C_32_H_27_O_11_N	III	R_1_ = OH; R_2_ = OH; R_3_ = H; R_4_ = Ac; R_5_ = H; R_6_ = OH	nothing	629.58	[8]
15	2″-*O*-Acetyl cyanidin	C_32_H_27_O_11_N_3_	III	R_1_ = OH; R_2_ = OH; R_3_ = Ac; R_4_ = H; R_5_ = H; R_6_ = OH	nothing	629.58	[8]
16	Quercetin 2″-*O*-(β-d-xyranosyl)-β-d-glucoside	C_26_H_28_O_15_	III	R_1_ = OH; R_2_ = H; R_3_ = D-xyl; R_4_ = H; R_5_ = H; R_6_ = OH	nothing	580.50	[8]
17	Quercetin 2″-*O*-(β-d-arabinopyranoside)	C_26_H_28_O_15_	III	R_1_ = OH; R_2_ = H; R_3_ = D-ara; R_4_ = H; R_5_ = H; R_6_ = OH	nothing	580.50	[8]
18	Rhamnetin 2″-*O*-β-l-rhamnoside	C_27_H_30_O_16_	III	R_1_ = OH; R_2_ = H; R_3_ = L-gal; R_4_ = H; R_5_ = H; R_6_ = OH	nothing	610.53	[8]
19	Kaempferol 2″-*O*-β-d-glucopyranoside	C_27_H_30_O_15_	III	R_1_ = OH; R_2_ = H; R_3_ = D-glu; R_4_ = H; R_5_ = H; R_6_ = OH	nothing	609.15	[8]
20	Kaempferol 6″-*O*-glucopyranoside	C_26_H_27_O_14_N	III	R_1_ = OH; R_2_ = H; R_3_ = H; R_4_ = H; R_5_ = D-glu; R_6_ = OH	nothing	577.50	[8]
21	6″-*O*-acetylkaempferol	C_32_H_27_O_10_N_3_	III	R_1_ = OH; R_2_ = H; R_3_ = H; R_4_ = H; R_5_ = Ac; R_6_ = OH	nothing	613.58	[8]
22	3″-*O*-acetylkaempferol	C_32_H_27_O_10_N_3_	III	R_1_ = OH; R_2_ = H; R_3_ = H; R_4_ = Ac; R_5_ = H; R_6_ = OH	nothing	613.58	[8]
23	2″-*O*-acetylkaempferol	C_32_H_27_O_10_N_3_	III	R_1_ = OH; R_2_ = H; R_3_ = Ac; R_4_ = H; R_5_ = H; R_6_ = OH	nothing	613.58	[8]
24	Genistein-7-*O*-β-d-pyranosylglucoside	C_22_H_22_O_10_	III	R_1_ = H; R_2_ = H; R_3_ = H; R_4_ = H; R_5_ = H; R_6_ = OCH_3_	nothing	446.41 284.26	[8]
25	3″-*O*-(2‴-methylbutanoyl)resveratrol	C_26_H_28_O_12_	IV	R_1_ = OH; R_2_ = OH	nothing	532.15	[8]
26	3″-*O*-(2‴-methylbutanoyl)quercetin	C_27_H_30_O_11_	IV	R_1_ = OCH_3_; R_2_ = H	nothing	530.52	[41]
27	3″-*O*-(2‴-methylbutanoyl) luteolin	C_26_H_28_O_11_	IV	R_1_ = OH; R_2_ = H	nothing	517.17	[42]
28	3″-*O*-(2‴-methylbutanoyl) chrysoeriol	C_27_H_30_O_12_	IV	R_1_ = OCH_3_; R_2_ = OH	nothing	547.18	[42]
29	Isoorientin	C_21_H_20_O_11_	V	nothing	28608-75-5	448.38	[40]
30	2″-*O*-feruloylharpagoside	C_24_H_31_O1_8_	VI	R_1_ = H; R_2_ = OH; R_2_ = OCH_3_	nothing	607.14	[43]
31	2″-*O*-feruloylverbascoside	C_31_H_27_O_14_	VI	R_1_ = OH; R_2_ = OH; R_3_ = OCH_3_	nothing	623.13	[43]
32	2″-*O*-feruloylisovitexin	C_31_H_27_O_11_	VI	R_1_ = H; R_2_ = CH_3_ O; R_3_ = OCH_3_	nothing	575.16	[44]
33	2″-*O*-(3‴-methoxycaffeoyl)luteolin	C_30_H_26_O_13_	VI	R_1_ = H; R_2_ = OH; R_3_ = OH	nothing	594.14	[44]
34	2″-*O*-feruloylgenistin	C_32_H_30_O_13_	VI	R_1_ = OH; R_2_ = CH_3_ O; R_3_ = OCH_3_	nothing	622.17	[8]
35	2″-*O*-(2‴-methylbutanoyl)quercetin	C_26_H_28_O_11_	VII	R_1_ = OH; R_2_ = H; R_3_ = OH; R_4_ = CH_2_ CH_3_	nothing	515.15	[45]
36	2″-*O*-(2‴-methylbutanoyl)kaempferol	C_26_H_28_O_12_	VII	R_1_ = OH; R_2_ = OH; R_3_ = OH; R_4_ = CH_2_ CH_3_	nothing	531.14	[45]
37	4′-methoxy-2”-*O*-(2‴-methylbutanoyl)luteolin	C_32_H_30_O_13_	VII	R_1_ = OH; R_2_ = H; R_3_ = OCH_3_; R_4_ = CH_2_CH_3_	nothing	623.17	[44]
38	4′-methoxy-2″-*O*-(2‴-methylbutanoyl)apigenin	C_32_H_30_O_14_	VII	R_1_ = OH; R_2_ = OH; R_3_ = OCH_3_; R_4_ = CH_2_CH_3_	nothing	639.17	[44]
39	2″-*O*-(2‴-methylbutanoyl)isogenistin	C_20_H_33_ O_17_	VII	R_1_ = OCH_3_; R_2_ = OH; R_3_ = OH; R_4_ = CH_2_CH_3_	nothing	545.17	[43]
40	2″-*O*-(2‴-methylbutanoyl)isokanamycin A	C_27_H_29_ O_11_	VII	R_1_ = OCH_3_; R_2_ = H; R_3_ = OH; R_4_ = CH_2_CH_3_	nothing	529.17	[43]
41	2″-*O*-isopropylbenzoyl-isokanamycin A	C_32_H_31_O_14_	VII	R_1_ = OCH_3_; R_2_ = OH; R_3_ = OH; R_4_ = CH_3_	nothing	639.17	[46]
42	3″-*O*-veratroyl orientin	C_29_H_26_O_14_	VIII	R = OH	nothing	599.13	[44]
43	3″-*O*-veratroyl vitexin	C_31_H_28_O_13_	VIII	R = H	nothing	608.56	[44]
44	2″-*O*-benzoylisorhamnetin	C_29_H_26_O_12_	IX	nothing	nothing	567.14	[44]
45	3″-*O*-Acetylquercetin	C_31_H_28_O_13_	X	nothing	nothing	609.16	[44]
46	2″-*O*-Vanilloylquercetin	C_29_H_26_O_14_	XI	R_1_ = OH; R_2_ = OH; R_3_ = OH; R_4_ = OH	nothing	599.13	[44]
47	2″-*O*-(3‴,4‴-dimethoxybenzoyl)isorhamnetin	C_31_H_30_O_14_	XI	R_1_ = OCH_3_; R_2_ = OH; R_3_ = OH; R_4_ = OCH_3_	nothing	626.16	[45]
48	2″-*O*-(3‴,4‴-dimethoxybenzoyl) isoswertisin	C_31_H_30_O_13_	XI	R_1_ = OCH_3_; R_2_ = H; R_3_ = OH; R_4_ = OCH_3_	nothing	610.17	[45]
49	2″-*O*-(3‴,4‴-dimethoxybenzoyl)isodaidzein	C_30_H_28_O_13_	XI	R_1_ = OH; R_2_ = H; R_3_ = OH; R_4_ = OCH_3_	nothing	595.14	[47]
50	2″-*O*-(3‴,4‴-dimethoxybenzoyl)quercetin	C_30_H_28_O_14_	XI	R_1_ = OH; R_2_ = OH; R_3_ = OH; R_4_ = OCH_3_	nothing	611.14	[47]
51	2″-*O*-vanilloylisorhamnetin	C_30_H_28_O_13_	XI	R_1_ = OCH_3_; R_2_ = H; R_3_ = OH; R4 = OH	nothing	597.16	[48]
52	2″-*O*-vanilloylquercetin	C_29_H_26_O_13_	XI	R_1_ = OH; R_2_ = H; R_3_ = OH; R_4_ = OH	nothing	582.5	[45]
53	Salvigenin	C_18_H_16_O_6_	XII	R_1_ = H; R_2_ = OCH_3_; R_3_ = OCH_3_; R_4_ = OH; R_5_ = H; R_6_ = H; R_7_ =OCH_3_; R_8_ = H	19103-54-9	328.31	[8]
54	Acacetin	C_16_H_12_O_5_	XII	R_1_ = H; R_2_ = OH; R_3_ = H; R_4_ = OH; R_5_ = H; R_6_ = H; R_7_ = OCH_3_; R_8_ = H	480-44-4	284.26	[8]
55	Apigenin	C_15_H_10_O_5_	XII	R_1_ = H; R_2_ = OH; R_3_ = H; R_4_ = OH; R_5_ = H; R_6_ = H; R_7_ = OH; R_8_ = H	520-36-5	270.24	[8]
56	Pectolinarigenin	C_17_H_14_O_6_	XII	R_1_ = H; R_2_ = OH; R_3_ = OCH_3_; R_4_ = OH; R_5_ = H; R_6_ = H; R_7_ = OCH_3_; R_8_ = H	520-12-7	314.29	[8]
57	Cirsimaritin	C_17_H_14_O_6_	XII	R_1_ = H; R_2_ = OCH_3_; R_3_ = OCH_3_; R_4_ = OH; R_5_ = H; R_6_ = H; R_7_ = OH; R_8_ = H	6601-62-3	314.29	[8]
58	Luteolin	C_15_H_10_O_6_	XII	R_1_ = H; R_2_ = OH; R_3_ = H; R_4_ = OH; R_5_ = H; R_6_ = OH; R_7_ = OH; R_8_ = H	491-70-3	286.24	[8]
59	Quercetin	C_15_H_10_O_7_	XII	R_1_ = H; R_2_ = OH; R_3_ = H; R_4_ = OH; R_5_ = OH; R_6_ = OH; R_7_ = OH; R_8_ = H	73123-10-1	302.23	[8]
60	Naringenin	C_15_H_12_O_5_	XII	R_1_ = H; R_2_ = OH; R_3_ = H; R_4_ = OH; R_5_ = H; R_6_ = H; R_7_ = OH; R_8_ = H	480-41-1	272.25	[8]
61	Chrysoeriol	C_16_H_12_O_6_	XII	R_1_ = H; R_2_ = OH; R_3_ = H; R_4_ = OH; R_5_ = H; R_6_ = OCH_3_; R_7_ = OH; R_8_ = H	491-71-4	300.26	[8]
62	Diosmetin	C_16_H_12_O_6_	XII	R_1_ = H; R_2_ = OH; R_3_ = H; R_4_ = OH; R_5_ = H; R_6_ = OH; R_7_ = OCH_3_; R_8_ = H	520-34-3	300.26	[8]
63	Farnisin	C_16_H_12_O_5_	XII	R_1_ = H; R_2_ = OH; R_3_ = H; R_4_ = H; R_5_ = H; R_6_ = OH; R_7_ = OCH_3_; R_8_ = H	54867-60-6	284.26	[8]
64	Kaempferol	C_15_H_10_O_6_	XII	R_1_ = H; R_2_ = OH; R_3_ = H; R_4_ = OH; R_5_ = OH; R_6_ = H; R_7_ = OH; R_8_ = H	520-18-3	286.24	[8]
65	Myricetin	C_15_H_10_O_8_	XII	R_1_ = H; R_2_ = OH; R_3_ = H; R_4_ = OH; R_5_ = OH; R_6_ = OH; R_7_ = OH; R_8_ = OH	529-44-2	318.23	[8]
66	Neodiosmin	C_28_H_32_O_15_	XII	R_1_ = H; R_2_ = O-rutinoside; R_3_ = H; R_4_ = OH; R_5_ = H; R_6_ = H; R_7_ = OCH_3_; R_8_ = H	38665-01-9	608.54	[8]
67	8-C-β-d-pyranosyl catechin	C_21_H_20_O_10_	XII	R_1_ = D-xyl; R_2_ = H; R_3_ = H; R_4_ = OH; R_5_ = H; R_6_ = OH; R_7_ = OCH_3_; R_8_ = H	nothing	432.38	[9]
68	7-*O*-viciafuranosyl quercetin	C_28_H_32_O_14_	XII	R_1_ = H; R_2_ = O-rutinoside; R_3_ = H; R_4_ = OH; R_5_ = H; R_6_ = OH; R_7_ = OCH_3_; R_8_ = H	nothing	593.19	[47]
69	7-*O*-naringenin rutinoside	C_28_H_32_O_14_	XII	R_1_ = H; R_2_ = O-neohesperidoside; R_3_ = H; R_4_ = OH; R_5_ = H; R_6_ = OH; R_7_ = OCH_3_; R_8_ = H	20633-93-6	607.16	[45]
70	Quercetin-3-*O*-β-l-rhamnoside	C_21_H_20_O_11_	XII	R_1_ = H; R_2_ = OH; R_3_ = H; R_4_ = OH; R_5_ = O-β-l-rhamnoside; R_6_ = OH; R_7_ = OH; R_8_ = H	522-12-3	448.38	[9]
71	Quercetin-3-*O*-β-d-glucopyranoside	C_21_H_20_O_11_	XII	R_1_ = H; R_2_ = OH; R_3_ = H; R_4_ = OH; R_6_ = OH; R_7_ = OH; R_5_ = O-β-d-glucopyrano- side; R_8_ = H	21637-25-2	448.37	[9]
72	5-Hydroxy-4′,7,8-trimethoxyflavone	C_18_H_16_O_6_	XII	R_1_ = OCH_3_; R_2_ = OCH_3_; R_3_ = H; R_4_ = OH; R_5_ = H; R_6_ = H; R_7_ = OCH_3_; R_8_ = H	57096-03-4	328.09	[8]
73	4′,5-Dihydroxy-7,8-dimethoxyflavone	C_17_H_14_O_6_	XII	R_1_ = OCH_3_; R_2_ = OCH_3_; R3 = H; R_4_ = OH; R_5_ = H; R_6_ = H; R_7_ = OH; R_8_ = H	6608-33-9	314.08	[8]
74	6‴-(3-hydroxy-3-methylbutanoyl)-2″-*O*-β-d-pyranosyl-hongcaoside	C_33_H_38_O_20_	XIII	R = OH	nothing	777.18	[12]
75	6‴-(3-hydroxy-3-methylbutanoyl)-2″-*O*-β-d-pyranosylmatrine	C_33_H_38_O_19_	XIII	R = H	nothing	761.19	[12]
76	2″-*O*-veratroylvitexin	C_30_H_28_O_13_	XIV	nothing	nothing	596.15	[8]
77	Isodaphnetin-2″-*O*-(6-*O*-feruloyl)-β-l-lactoside	C_37_H_38_O_18_	XV	nothing	nothing	771.21	[49]
78	Hyperoside	C_21_H_20_O_12_	XVI	R_1_ = OH; R_2_ = OH; R_3_ = CH_2_OH	482-36-0	464.40	[8]
79	Naringenin 3-(6″-ethyl glucuronide)	C_23_H_24_O_10_	XVI	R_1_ = OH; R_2_ = H; R_3_ = COOCH_2_CH_3_	nothing	460.14	[8]
80	Astragalin	C_21_H_20_O_11_	XVI	R_1_ = OH; R_2_ = H; R_3_ = CH_2_OH	480-10-4	448.40	[8]
81	Eriodictyol	C_15_H_12_O_6_	XVII	nothing	552-58-9	288.25	[8]
82	2″-*O*-(2‴-*O*-methybutyryl)-glucopyranosyl isoswertisin	C_33_H_40_O_16_	XVIII	nothing	nothing	691.22	[50]
83	2″-*O*-(6‴-*O*-veratroyl)-galactopyranosyl vitexin	C_36_H_38_O_18_	XIX	nothing	nothing	759.21	[50]
84	2″-*O*-(6‴-*O*-caffeoyl)-galactopyranosyl vitexin	C_36_H_36_O_18_	XX	R_1_ = H; R_2_ = OH; R_3_ = H; R_4_ = OH	nothing	757.19	[50]
85	2″-*O*-(6‴-*O*-feruloyl)-galactopyranosyl orientin	C_37_H_38_O_19_	XX	R_1_ = OH; R_2_ = OCH_3_; R_3_ = H; R_4_ = OH	nothing	787.20	[50]
86	Trollichinenside A(3″-*O*-veratroylvitexin)	C_36_H_35_O_18_	XX	R_1_ = OH; R_2_ = OH; R_3_ = H; R_4_ = OH	nothing	755.18	[8]
87	Trollichinenside B (3″-*O*- feruloylvitexin)	C_38_H_40_O_20_	XX	R_1_ = OH; R_2_ = OCH_3_; R_3_ = OCH_3_; R_4_ = OH	nothing	816.21	[8]
88	Trollichinenside C (6″-*O*-veratroylvitexin)	C_38_H_40_O_19_	XX	R_1_ = OH; R_2_ = H; R_3_ = OCH_3_; R_4_ = OCH_3_	nothing	800.22	[8]
89	Daidzin	C_21_H_20_O_9_	XXI	R_1_ = H; R_2_ = OH	552-66-9	416.41	[8]
90	Kaempferol-7-*O*-β-d-glucoside	C_22_H_22_O_10_	XXI	R_1_ = OH; R_2_ = OCH_3_	nothing	446.12	[8]
91	Glucosylorientin	C_27_H_30_O_17_	XXII	nothing	76135-83-6	626.5	[8]
92	6″-Malonylcosmosiin	C_24_H_22_O_13_	XXIII	nothing	86546-87-4	518.4	[8]
93	(-)-Gallocatechi	C_15_H_14_O_7_	XXIV	nothing	nothing	306.27	[8]
94	Quercetin-2″-*O*-β-l-arabinopyranoside	C_27_H_30_O_16_	XXV	nothing	861691-37-4	610.15	[8]
95	Apigenin-8-C-(2″-*O*-feruloyl)-β-d-glucoside	C_31_H_28_O_13_	XXVI	nothing	nothing	608.15	[8]
96	(2E)-2-methyl-1-*O*-vanilloyl-4-β-d-glucopyrano-side-2-butene	C_19_H_26_O_10_	XXVII	nothing	nothing	437.14	[46]
97	Neocarlinoside	C_26_H_28_O_15_	XXVIII	nothing	83151-89-7	580.5	[8]
98	4′,5-dihydroxy-3′,7-dimethoxy-isoflavone	C_17_H_14_O_6_	XXIX	R_1_ = OCH_3_; R_2_ = H; R_3_ = OH; R_4_ = OCH_3_	nothing	314.08	[8]
99	Glycitein	C_16_H_12_O_5_	XXIX	R_1_ = OH; R_2_ = OCH_3;_ R_3_ = H; R_4_ = H	40957-83-3	284.26	[8]
100	Daidzein	C_15_H_10_O_4_	XXIX	R_1_ = OH; R_2_ = H; R_3_ = H; R_4_ = H	486-66-8	254.23	[8]

**Table 4 molecules-29-00421-t004:** Phenolic acids isolated from *T. chinensis*.

No	Names	Molecular Formula	Parent Nucleus	Substituent	CAS	Molecular Weight	Refs.
101	Trollioside	C_19_H_26_O_9_	XXX	R_1_ = O-β-d-glucopyranosyl; R_2_ = OCH_3_	nothing	399.16	[9]
102	Proglobellowery acid	C_7_H_6_O_2_	XXX	R_1_ = OH; R_2_ = OCH_3_	nothing	235.00	[9]
103	4-(β-d-glucopyranosyloxy)-3- (3-methyl-2-butenyl)benzoic acid	C_18_H_24_O_8_	XXX	R_1_ = O-β-d-glucopyranosyl; R_2_ = H	nothing	368.38	[9]
104	4-Hydroxybenzoic acid	C_7_H_6_O_3_	XXXI	R_1_ = COOH; R_2_ = H; R_3_ = OH; R_4_ = H	99-96-7	138.03	[9]
105	3,4-dihydroxybenzoic acid methyl ester	C_8_H_8_O_2_	XXXI	R_1_ = COOCH_3_; R_2_ = OH; R_3_ = OH; R_4_ = H	2150-43-8	152.05	[9]
106	Methylparaben	C_6_H_4_O_3_	XXXI	R_1_ = COOCH_3_; R_2_ = H; R_3_ = OH; R_4_ = H	35816-31-0	152.05	[9]
107	Protocatechuic acid	C_7_H_6_O_4_	XXXI	R_1_ = COOH; R_2_ = OH; R_3_ = OH; R_4_ = H	99-50-3	154.12	[9]
108	Methyl veratrate	C_10_H_12_O_4_	XXXI	R_1_ = COOCH_3_; R_2_ = OCH_3_; R_3_ = OCH_3_; R_4_ = H	2150-38-1	196.20	[9]
109	Benzoic acid	C_7_H_6_O_2_	XXXI	R_1_ = COOH; R_2_ = H; R_3_ = H; R_4_ = H	117500-35-3	122.12	[9]
110	Veratric acid	C_9_H_10_O_4_	XXXI	R_1_ = COOH; R_2_ = OCH_3_; R_3_ = OCH_3_; R_4_ = H	93-07-2	182.17	[9]
111	Vanillic acid	C_8_H_8_O_4_	XXXI	R_1_ = COOH; R_2_ = OCH_3_; R_3_ = OH; R_4_ = H	121-34-6	168.14	[8]
112	Gallic acid	C_7_H_6_O_5_	XXXI	R_1_ = COOH; R_2_ = OH; R_3_ = OH; R_4_ = OH	149-91-7	170.12	[8]
113	4-Hydroxy-2,6-dimethoxybenzaldehyde	C_9_H_10_O_4_	XXXI	R_1_ = CHO; R_2_ = OCH_3_; R_3_ = OH; R_4_ = OCH_3_	22080-96-2	182.17	[8]
114	Monotropein	C_16_H_22_O_11_	XXXII	nothing	5945-50-6	390.33	[8]
115	Ascorbic acid	C_6_H_8_O_6_	XXXIII	nothing	299-36-5	176.13	[8]
116	3-(6-hydroxy-7-methoxy-2H-1,3-benzodioxol-5-yl)propanoic acid	C_11_H_12_O_6_	XXXIV	nothing	nothing	240.06	[8]
117	(2R,3S)-Piscidic acid	C_11_H_12_O_7_	XXXV	nothing	469-65-8	256.06	[8]
118	Isochlorogenic acid A	C_25_H_24_O_12_	XXXVI	nothing	2450-53-5	516.46	[8]
119	Tecomin	C_15_H_20_O_9_	XXXVII	nothing	31002-27-4	344.31	[8]
120	Shikimic acid	C_7_H_10_O_5_	XXXVIII	nothing	138-59-0	174.15	[8]
121	Phlorizin dihydrate	C_21_H_26_O_11_	XXXIX	nothing	7061-54-3	454.43	[8]
122	Globeflowery acid	C_13_H_16_O_4_	XL	nothing	4041-28-5	236.26	[8]
123	Ursolic acid	C_30_H_48_O_3_	XLI	nothing	77-52-1	456.71	[8]
124	Salviaflaside	C_24_H_26_O_13_	XLII	nothing	178895-25-5	522.46	[8]
125	Rhynchophylline	C_14_H_6_O_8_	XLIII	nothing	76-66-4	302.19	[8]
126	Methyl dodecanoate	C_13_H_26_O_2_	XLIV	R = H	111-82-0	214.34	[53]
127	Methyl tridecanoate	C_14_H_28_0_2_	XLIV	R = CH_2_CH_3_	1731-88-0	228.37	[53]
128	Methyl tetradecanoate	C_15_H_30_O_2_	XLIV	R = (CH_2_)_3_CH_3_	124-10-7	242.40	[53]
129	Methyl pentadecanoate	C_16_H_32_O_2_	XLIV	R = (CH_2_)_4_CH_3_	7132-64-1	256.42	[53]
130	Methyl hexadecanoate	C_17_H_34_O_2_	XLIV	R = (CH_2_)_7_CH	112-39-0	270.45	[53]
131	Methyl heptadecanoate	C_18_H_36_O_2_	XLIV	R = (CH_2_)_8_CH_3_	1731-92-6	284.47	[53]
132	Methyl octadecanoate	C_19_H_38_O_2_	XLIV	R = (CH_2_)_9_CH_3_	112-61-8	298.50	[53]
133	Methyl eicosanoate	C_21_H_42_O_2_	XLIV	R = (CH_2_)_11_CH_3_	22589-04-4	326.55	[53]
134	Methyl docosanoate	C_23_H_46_O_2_	XLIV	R = (CH_2_)_13_CH_3_	929-77-1	354.61	[53]
135	Methyl tetracosanoate	C_25_H_50_O_2_	XLIV	R = (CH_2_)_15_CH_3_	2442-49-1	382.66	[53]
136	Methyl decanoate	C_11_H_22_O_2_	XLIV	R = CH_2_CH_3_	110-42-9	186.29	[53]
137	Methyl octanoate	C_9_H_18_O_2_	XLIV	R = H	111-11-5	158.23	[53]
138	Methyl benzeneacetate	C_9_H_10_O_2_	XLV	nothing	143390-89-0	150.17	[53]
139	Methyl benzoate	C_8_H_8_O_2_	XLVI	R_1_ = H, R_2_ = H	36712-21-7	136.15	[53]
140	3,4-dimethoxybenzoic acid methyl ester	C_10_H_12_O_4_	XLVI	R_1_ = OCH_3_; R_2_ = OCH_3_	2150-38-1	196.2	[53]
141	Dimethyl 3-hydroxy-2-methyl-glutarate	C_8_H_14_0_5_	XLVII	nothing	nothing	190.19	[53]
142	N-hexadecanoic acid	C_16_H_32_O_2_	XLVIII	R_1_ = H; R_2_ = H	57-10-3	256.42	[53]
143	Hexadecanoic acid,10,16-dihydroxy	C_16_H_32_O_4_	XLVIII	R_1_ = OH; R_2_ = OH	3233-90-7	288.42	[53]
144	Dimethyl butanedioate	C_6_H_10_O_4_	XLIX	nothing	106-65-0	146.14	[53]
145	Dimethyl octanedioate	C_10_H_18_O_4_	L	nothing	1732-09-8	202.24	[53]
146	Dimethyl nonanedioate	C_11_H_26_O_4_	LI	nothing	1732-10-1	216.27	[53]
147	2-hydroxyhexadecanoic acid methyl ester	C_17_H_34_O_3_	LII	R_1_ = OH; R_2_ = H; R_3_ = H	78330-57-1	286.45	[53]
148	3-hydroxyhexadecanoic acid methyl ester	C_17_H_34_O_3_	LII	R_1_ = H; R_2_ = OH; R_3_ = H	51883-36-4	286.45	[53]
149	10-hydroxyhexadecanoic acid methyl ester	C_17_H_34_O_3_	LII	R_1_ = H; R_2_ = H; R_3_ = OH	56247-30-4	286.45	[53]
150	2-hydroxy-benzaldehyde oxime	C_7_H_7_NO_2_	LIII	nothing	94-67-7	137.14	[53]
151	4-hydroxy-acetophenone	C_8_H_8_O_2_	LIV	nothing	99-93-4	136.15	[53]
152	2-methoxydocosyl methanoate	C_25_H_50_O_3_	LV	nothing	nothing	398.66	[53]
153	3-phenylprop-2-enoic acid methyl ester	C_11_H_12_O	LVI	R_1_ = H; R_2_ = H	103-26-4	160.00	[53]
154	3-(4-hydroxyphenyl)prop-2-enoic acid methyl ester	C_10_H_10_O_3_	LVI	R_1_ = H; R_2_ = OH	61240-27-5	178.00	[53]
155	(4-hydroxy-3-methoxyphenyl)-2-propenoic acid methyl ester	C_11_H_12_O_4_	LVI	R_1_ = OCH_3_, R_2_ = OH	34298-89-0	208.00	[53]
156	2,3-Dihydrobenzofuran	C_8_H_8_O	LVII	nothing	496-16-2	120.15	[53]
157	4-phenyl-2-butenoic acid methyl ester	C_11_H_12_O	LVIII	nothing	54966-43-7	176	[53]
158	9-(propoxybenzene)-nonanoic acid methyl ester	C_19_H_30_O_2_	LIX	nothing	nothing	290	[53]
159	(E)-11-eicosenoic acid methyl ester	C_21_H_40_O_2_	LX	nothing	nothing	324	[53]
160	(Z)-9-hexadecenoic acid methyl ester	C_17_H_32_O_2_	LXI	R = (CH_2_)_4_CH_3_	1120-25-8	268.43	[53]
161	(Z)-9-octadecenoic acid methyl ester	C_19_H_36_O_2_	LXI	R = (CH_2_)_6_CH_3_	112-62-9	296.48	[53]
162	(Z,Z)-9,12-octadecadienoic acid methyl ester	C_19_H_32_O_4_	LXI	R = CH = CH(CH_2_)_4_CH_3_	168482-44-8	294.47	[53]
163	(Z,Z,Z)-9,12,15-Octadecatrienoic acid methyl ester	C_19_H_32_O_2_	LXI	R = CH = CHCH_2_CH = CHCH_2_CH_3_	301-00-8	292.46	[53]
164	(Z,Z,Z)-9,12,15-ctadecatrien-1-ol	C_18_H_32_O	LXII	nothing	nothing	264	[53]

**Table 5 molecules-29-00421-t005:** Alkaloids isolated from *T. chinensis*.

No	Names	Molecular Formula	Parent Nucleus	Substituent	CAS	Molecular Weight	Refs.
165	Senecionine	C_18_H_25_NO_5_	LXIII	Nothing	130-01-8	335.4	[56]
166	Integerrimine	C_18_H_25_NO_5_	LXIV	Nothing	480-79-5	335.4	[46]
167	Trolline	C_12_H_13_NO_3_	LXV	Nothing	1021950-79-7	219.24	[9]
168	(R)-Cyanomethyl-3-hydroxyindole	C_10_H_7_O_2_N_2_	LXVI	Nothing	Nothing	187.05	[46]
169	Adenine	C_5_H_5_N_5_	LXVII	Nothing	73-24-5	135.13	[8]

**Table 6 molecules-29-00421-t006:** Other chemical components isolated from *T. chinensis*.

No	Names	Molecular Formula	Parent Nucleus	Substituent	Characterization Method	Molecular Weight	Refs.
168	Daucosterol	C_35_H_60_O_6_	LXVIIII	nothing	474-58-8	576.85	[8]
169	Trolliusol A	C_17_H_16_O_6_	LXIX	nothing	nothing	316.30	[8]
170	Esculetin	C_9_H_6_O_4_	LXX	nothing	305-01-1	178.14	[8]
171	β-Sitosterol	C_29_H_50_O	LXXI	nothing	5779-62-4	414.71	[8]
172	Trolliamide	C_42_H_82_NO_5_	LXXII	nothing	nothing	680.62	[11]
173	L-Rhamnose	C_6_H_14_O_6_	LXXIII	nothing	6155-35-7	182.17	[8]
174	L-Arabinose	C_5_H_10_O_5_	LXXIV	nothing	5328-37-0	150.13	[8]
175	D-Galactose	C_6_H_12_O_6_	LXXV	nothing	59-23-4	180.15	[8]
176	Vanillylamine	C_8_H_11_NO_2_	LXXVI	nothing	1196-92-5	153.18	[8]
177	2-(3,4-Dihydroxyphenyl)ethyl-*O*-β-d-pyranoglucose	C_8_H_10_O_3_	LXXVII	R = H	10597-60-1	154.16	[8]
178	Homovanillyl alcohol	C_9_H_12_O_3_	LXXVIII	nothing	2380-78-1	168.19	[8]
179	2-(3,4-dihydroxyphenyl)-ethyl-*O*-β-d-glucopyranoside	C_14_H_20_O_8_	LXXIX	nothing	nothing	315.10	[8]
180	3,5-dihydroxyphenethyl alcohol 3-0-β-d-glucopyranoside	C_14_H_20_O_8_	LXXX	nothing	52674-86-9	315.10	[8]
181	4′-*O*-(6″-*O*-Vanilloylajugol-β-d-glucopyranoyl)phenylethanol	C_22_H_26_O_10_	LXXXI	nothing	27606-08-2	450.44	[8]
182	Xantho-phyll-epoxyde	C_40_H_56_O_3_	LXXXII	nothing	nothing	584.87	[64]
183	Trollixanthin	C_40_H_56_N_4_	LXXXIII	nothing	14660-91-4	592.90	[64]

**Table 7 molecules-29-00421-t007:** Main pharmacological effects of TC.

Pharmacological Effects	Extracts/Compounds	Animals/Cells	Dosage/Concentration	Effects/Mechanisms	References
Antiviral	The crude extract from the flowers of *T. chinensis*	ICR mice	0.2 mg/g/d	The *T. chinensis* crude extract treatment resulted in a significant increase in the body weight percentage, a decrease in the number of white blood cells, and a lowered lung index among mice infected with influenza virus A/FM/1/47 (H1N1) virus.	[115]
	Orientin	Hep 2 cell	0.1 mL of maintenance medium containing serial two-fold dilutions of the tested compounds 0.1 mL of maintenance medium without the test compound was added	The flavonoids isolated from *T. chinensis*, Orientin, and Vitexin, possess strong anti-viral activities against Para 3. Proglobeflowery acid showed weak antiviral activity against Para 3.	[10]
	Vitexin				
	Proglobeflowery acid				
	The crude extract from the flowers of *T. chinensis*	ICR mice	0.2 mg/g/d	The crude extract from the flowers of *T. chinensis* was found to inhibit the increased expression of TLR3, TBK1, TAK1, and IRF3 induced by the high-dose influenza virus and treat mice infected with influenza virus by activating the TLR3 signaling pathway.	[1]
	Veratric acid	RAW264.7 cell	50, 100, 200, 400, and 800 μmol/L	The three representative compounds play a role in anti-H1N1 viral effects by regulating the TLR 3, 4, and 7 pathways, counteracting the inflammatory damage caused by excessive production of NO, IL-1, IL-6, and TNF induced by viral infection, and promoting the production of IFN- to eliminate the virus.	[116]
	Vitexin				
	Trolline				
	Piscidic acid	EV71-infected RD cells	The *T. chinensis* mother liquor was diluted by a factor of 2^0^, 2^−1^, 2^−2^, 2^−3^, 2^−4^, and 100 μL was dispensed into each well of the cell culture plate	The viral inhibition rate of *T. chinensis* ranges from 49.64% to 73.69%. It exhibits a clear inhibitory effect on the EV71 virus, and the three compounds form the foundation of *T. chinensis*’ anti-EV71 material.	[25]
	2″-Oacetylorientin				
	2-(4-hydroxybenzyl) malic acid				
Antioxidant	Orientin	Not applicable	46/5.64/5.19/3.97 mg/mL	Under in vitro conditions, phenolic and flavonoid compounds could efficiently scavenge a variety of ROS or DPPH-free radicals.	[117]
	Vitexin				
	Proglobeflowery acid				
	Orientin	KM mice	40/20/10 mg/kg	Slowing down d-galactose-induced aging by enhancing the activity of antioxidant enzymes, eliminating excessive oxygen free radicals, and mitigating damage to cells and tissues.	[118]
	Vitexin				
Antiinflammatory	Trolliusditerpenosides A-Q (1–17)	RAW 264.7 cells mediated by LPS	Not applicable	The inhibitory effects on LPS-induced NO (pro-inflammatory mediator nitric oxide) release in RAW 264.7 cells by diterpenoid glycosides from *T. chinensis.*	[14]
	Orientin	RAW264.7 cells	0, 25, 50, 100, 200, and 400 µmol/L^−1^	The production of NO, IL-6, and TNF-stimulated cells decreased.	[119]
	Vitexin				
	Quercetin				
	Isoquercetin				
	Veratric acid				
	Proglobeflowery acid				
	Trollioside				
	2″-*O*-β-l-galactopyranosy-orientin				
	Luteolin	RAW264.7 cells	0, 12.5, 25, 50, 100, 200, and 400 µmol·L^−1^	The production of NO, IL-6, and TNF-stimulated cells decreased.	[119]
	Trolline				
	Aqueous extract of the stem and leaves of *T. chinensis*	KM mice	Distilled water: 20 mL/kg Positive drug: 100 mg/kg Aqueous extract of stem and leaves of *T. chinensis* (low/high): 12 g/kg/24 g/kg Alcohol extract of the stem and leaves of *T. chinensis* (low/high): 12 g/kg/24 g/kg	*T. chinensis* has some anti-inflammatory effects on stem and leaf extracts.	[111]
	Alcohol extract of the stem and leaves of *T. chinensis*				
Antitumor	Total flavonoids	MCF-7 cells	0/0.0991/0.1982/0.3964/0.7928/1.5856 mg/mL	Flavonoids were found to suppress growth and induce apoptosis in MCF-7 cells.	[93]
	Orientin	EC-109 cells	5.0, 10.0, 20.0, 40.0, and 80.0 µM	Orientalin and Vitexin reduce apoptosis in human esophageal cancer EC-109 cells by regulating oncogenes and tumorigenic genes.	[120]
	Vitexin				
Antibacterial	Trolliusol A	*M. albicans* *E. coli* *P. aeruginosa* *B. subtilis* *S. aureus*	Drug concentrations: 1:4, 1:8, 1:16, 1:32, 1:64, 1:128, 1:256, 1:512, 1:1024, 1:2048, and 1:4096	Minimal Inhibitory Concentration (MIC) was achieved by the microbroth method to achieve inhibition efficiency.	[41]
	1-(3′,4′-dihydroxyphenyl)-6,7-dihydroxyisochroman				
	(S)-1-(3′,4′-dihydroxyphenyl)-1-hydroxypropan-2-one				
	3,4-dihydroxyphenylethanol				
	2″-*O*-(2‴-methylbutyryl)isoswertisin				
	3″-*O*-(2‴-methylbutyryl)isoswertisin				
	Isoswertisin				
	Orientin				
	Water extracts from *T. chinensis*	Microorganism *S. mutans*	50/25/12.5/6.25/3.125 (mg/mL)	*T. chinensis* has antibacterial and anti-inflammatory effects and can be used against mutant baculoviruses. Thirty percent ethanol extractexhibited the best antibacterial and antibiofilm effects.	[121]
	30% ethanol extracts from*T. chinensis*				
	60% ethanol extracts from *T. chinensis*				
	90% ethanol extracts from *T. chinensis*				
Antiaging	Orientin	KM mice	40/20/10 mg/kg	It can enhance the activity of antioxidant enzymes, eliminate excessive oxygen-free radicals, and reduce the damage to cells and tissues so as to delay the senescence caused by D-galactose.	[118]
	Vitexin				
Antipyretic	Flavonoids	New Zealand rabbits	Flavonoids 200 mg·kg^−1^ group Flavonoids 100 mg·kg^−1^ group Flavonoids 50 mg·kg^−1^ group Asprin 100 mg·kg^−1^ group	By inhibiting the expression of TNF-α and IL-1β in serum and PGE2 in cerebrospinal fluid.	[110]
Analgesic	Aqueous extract of stem and leaves of *T. chinensis*	KM mice	Distilled water: 20 mL/kg Positive drug: 100 mg/kg Aqueous extract of stem and leaves of *T. chinensis* (low/high): 12 g/kg/24 g/kg Alcohol extract of the stem and leaves of *T. chinensis* (low/high): 12 g/kg/24 g/kg	*T. chinensis* extracts from stems and leaves have been shown to have some analgesic effects.	[111]
	Alcohol extract of stem and leaves of *T. chinensis*				
	The total flavones in *T. chinensis*	KM mice	125, 250, and 2500 mg/kg	It may increase the pain threshold of the hot plate in mice and have analgesic effects.	[112]
Antitussive and Expectorant	The total flavones in *T. chinensis*	KM mice	125, 2250, and 2500 mg/kg	The total flavonoid extract of *T. chinensis* has obvious anti-tussive and expectorant effects.	[112]
Myocardial ischemia/reperfusion injury (MI/RI)	The total flavones in *T. chinensis*	SD rats	50 mg/(kg·d)–100 mg/(kg·d)	The total flavones in *T. chinensis* protect the myocardium from MI/RI.	[113]

Not applicable means not described in detail in the literature.

## Data Availability

No data were used for the research described in the article.

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
