# Peer review of "Trollius chinensis Bunge: A Comprehensive Review of Research on Botany, Materia Medica, Ethnopharmacological Use, Phytochemistry, Pharmacology, and Quality Control"

_molecules, 2024, doi:10.3390/molecules29020421_

Round 1

Reviewer 1 Report

Comments and Suggestions for Authors

It is an extensive manuscript that compiles important information about a plant species, Trollius chinensis, which is important in traditional Chinese medicine.

However, the document has serious deficiencies in integrating and presenting information. The writing and spelling errors must be greatly improved. The file is sent with some annotations.

Reviewer 2 Report

Comments and Suggestions for Authors

Dear Authors

First of all I would like to congratulate you for putting such outstanding information in such an excellent way together.

Your paper summarises and explores materia medica, traditional uses, botany, pharmaceutical chemical composition, pharmacology effects, and quality control on T. chinensis considering relevant literature from 1991 to 2023. Even the paper is excellent, some minor points may be considered for further improvement

Comments:

Material and methods: It would be nice to provide a little bit more detailed information of the cited data bases (accessibility, data included, search options)

Botany: Table of a total of 26 species of Trollius species classified according Latin name, distribution area and altitude. The reader may be also interested in the type of differences beside altitude, also impact of content and composition would be of interest. In an perfect world also some pictures could be of some interest to show relevant differences), if available. It is the first revie article bringing all these information together.

Research on materia medica: It is a very nice overview of the different appearance of these species from middle ages up to now. However, this section is difficult to read. Maybe a revision to provide a more structured description (from oldest to newest descriptions) is helpful.

Is there any reason known why T. chinensis has no longer been documented in the Chinese Pharmacopoeia? If yes, a short comment would be very helpful in particular for those readers not familiar with the Chinese Monograph.

The traditional and current use are nicely presented and additionally summarized in a table. Some comments why the use has been changed over time could be very useful.

The table summarizing the chemical composition isolated from Trollius chinensis Bge. Is very impressive. It would be great to guide the reader at the beginning to the listed parent nuclei shown after the table.

Knowing it may be not possible to include known pharmacological activities to the single components (at least the ones where some data are reported, it would be great to have such a compilation. It would have a huge added value. Also CAS numbers where applicable could be a great ad on.

The parent nuclei structures have different sizes. It would be great to apply the same drawing style to all structures including the stereochemistry (where applicable). The sugar moieties are drawn in the chair conformation, a more consequent 2D-structure could be considered. Anyway, a unification of the drawings is essential for the overall quality.

The section of pharmacological effects is very nicely structured and complemented by a very nice overview table.

In figure 4 the abbreviations for the targets should be explained as footnotes, even they are shown in the glossary to get the complete information of the content

Reviewer 3 Report

Comments and Suggestions for Authors

1.     Page 1, In “Introduction” section, you can write either genus Trollius (preferred here) or Trollius spp, while “genus Trollius spp” is a meaningless combination.

2.     The record of T. Chinensis in Supplements to Compendium of Materia Medica needs to cite the reference.

3.     Page 2, in the introduction section, it says “many pharmaceutical effects such as ….” has been demonstrated by T. Chinensis at the first sentence, and then third last sentences, again, it gives “T. Chinensis clearly demonstrated the same effects such as….”.    Please pay attention to the similar issues (that the same contents are frequently repeated) in the rest of manuscript, and make the text succinct.   

4.     Page 2, botany section, T. Chinensis is one species of Trollius genus, how could 26 species come from T. Chinensis?!

5.     Page 3, what does this mean?  “the main source was Ranunculaceae of T. chinensis, or Asian T. chinensis“?

6.     Page 4, “Leaves are pentagonal, pentagonal, pentagonal ovate,…”  ??

7.     Page 4, please refine the grammar of the content about plant morphology.

8.     Page 5, figure 2, what is the caption of the figure of Mongolia map?  Why is this map magnified?

9.      Page6, these are poorly translated name of the medicines. (JinLianHuaPian, JinLianHuaRunHouPian, Jin-LianHuaKouFuYe, JinLianHuaJiaoNang, and JinLianHuaKeLi)

10.  Page 8, the phytochemials’ names (e.g., carboside, oxyside, hexosaccharide) are poorly written and not in English, please check and ensure their source and reliability.          

11.  Page 8, it is impossible that the flavonoids are metabolized into carotenoids (Trollixanthin and xanthophyll-epoxide)

12.  Page 8, the last sentence of section 6.1, I don’t understand the correlation between almond and total flavonoid content of T. chinensis.

13.  Please check all the chemical names in this manuscript, there are too many errors in it to be pointed out one by one.

14.  Page 9, please pay attention to the classification of phytochemicals: resins and volatile oils are not the names of any specific chemical category.

15.  Page 9, T. chinensis comprises of various monosaccharides, which has nothing to do with the polysaccharides and their activities as demonstrated.

16.  What is ultraviolet activity

17.  Figure 4, the effects and therapeutic targets/objectives of T. chinensis are illustrated in a very unclear or even misleading way. Please considering remaking this figure with more details.

18.  Page 36, it seems meaningless when you mentioned two categories of quality evaluation “the content determination methods of chemical composition analysis and the quality evaluation methods based on various fingerprinting techniques”, in which the quality evaluation method of T. chinensis based on fingerprinting techniques is using chemical content determination method

19.  Please carefully check the language and grammar of this manuscript, at least using some affordable software such as Grammarly to correct the language errors before submitting the revised version.

Comments on the Quality of English Language

No way to publish this manuscript to any reputed Journals under current language quality.   

Reviewer 4 Report

Comments and Suggestions for Authors

The review “Trollius chinensis Bunge : A Comprehensive Review of research on botany, materia medica,ethnopharmacological use, phytochemistry, pharmacology, and quality control” describes the progress of scientific understanding of T. chinensis in the area of botany, material medica, ethnopharmacological use, phytochemistry, pharmacology, and quality control. The review covers the literature of a long period (more that 30 years) from 1991 to 2023. The manuscript is well and carefully written, data are solid and conclusions are justified by the results. This manuscript could be accepted after minor revision. In the section 2 (material and methods) the authors are asked to add the key words while searching the literature data and the number of references that they have analyzed. Section 7.5 (and Figure 4): please make the names of bacteria in italic (as well as “in vitro” through the text).

Author Response

Dear reviewer,

Thank you very much for your comments and professional advice. These opinions help to improve the academic rigor of our article "Trollius chinensis Bunge: A Comprehensive Review of research on botany, materia medica, ethnopharmacological use, phytochemistry, pharmacology, and quality control(molecules-2774709)". We have corrected the revised manuscript's modifications based on your suggestion and request. We've highlighted them in red throughout the text.  Furthermore, we would like to show the details as follows:

Reviewer 4#

  1. In section 2 (material and methods) the authors are asked to add the key words while searching the literature data and the number of references they have analyzed.

The author's answer:

Thanks for your professional suggestions. For now, I've summarized your recommendations and added them to Section 2 (Materials and Methods) of the article, highlighted in red.

  1. Section 7.5 (and Figure 4): please make the names of bacteria in italics (as well as "in vitro" through the text).

The author's answer:

Thank you for your advice. As you suggested, we have italicized Section 7.5 (and Figure 4) of the names of bacteria and "in vitro" throughout the text.

Thank you again for your positive comments and valuable suggestions to improve the quality of our manuscript. Thank you very much for your attention and time. The new year is just around the corner, and we wish you all the best and a happy new year.

We look forward to hearing from you.

Yours sincerely,

Lian-Qing He

Name: Xiu-Bo Liu

Name: Wei-Chao Ren

Round 2

Reviewer 1 Report

Comments and Suggestions for Authors

Check my suggestion in the annex archive

Reviewer 3 Report

Comments and Suggestions for Authors

Authors have done some work on improving the quality of this manuscript but it still contains a plethora of issues that cannot be fixed by reviewers during current reviewing process.  Therefore, I confirm the recommendation of a rejection to this manuscript.

I give some examples of issues below.

1.       The remained issues from previous comments:

The issues mentioned in previous comments were partially fixed while others retain. For example, the misuse of Trollius as the species term. In the title of table 1, “a total of 26 species of Trollius species” should be “a total of 26 species of Trollius genus”. Another example is about the plant morphology: “Seeds are subobovoid, ca. 1-1.5 mm long, black, glossy. Fl. Jun--Jul, fr. Aug--Sep”.  “Fl.” and “Fr.” Probably mean flowering at... and fruiting at…. However, authors cannot expect every reader to understand them without a explanation.      Lots of wrong use of terms and words are still waiting to be corrected.

2.       Language quality:

I believe the authors have tried to improve the language of the manuscript. However, the general language quality of this manuscript stays in a very low level without sufficient improvement.  For example, in Abstract, authors mention a general symptom, oral sore throat which should be either “oral/throat sore”, “mouth/throat sore” or “sore mouth/throat”. Furthermore, this problematic term occurs in an incomplete sentence: “Its therapeutic process of clearing heat and detoxification, treating oral sore throat, earache, eye pain, sore throat, fever from a cold, and improving vision.”!    

Still in the Abstract section, authors state that: “Flavonoids, phenolic acids, and alkaloids were the major chemical constituents of T. chinensis”.  These phytochemicals are usually known in minor quantity which can be hardly considered as major chemical constituents compared to carbohydrate/lipid/protein of a wild plant. I believe authors know the correct answers for such issues, which, however, can hardly revised according to current endeavors from authors.   Plenty of these errors exist in the rest of the manuscript.  Rewriting these sentences is necessary but already out of scope of reviewers’ ability and responsibility.

3.       Apparent scientific or format errors:

For example, in figures/tables 3 and 4, it says “Flavones’ Parent nucleus isolated from T. chinensis” and “Phenolic acids’ Parent nucleus isolated from T. chinensis”. However, in these figures, plenty of parent structures are not the compounds as claimed.  Besides, in table 7, the study giving a dosage of 12-24g extract/kg model animals was wrongly cited.

There is no title for the table in page 22. If it is considered the continued content of table 5, then again, it is a mistitled table.

Comments on the Quality of English Language

Low
